# Spatial variability of snow precipitation and accumulation in COSMO–WRF simulations and radar estimations over complex terrain

Franziska Gerber[1,2], Nikola Besic[3,4], Varun Sharma[1], Rebecca Mott[2,5], Megan Daniels[6], Marco Gabella[4], Alexis Berne[3], Urs Germann[4], and Michael Lehning[1,2]

[1]Laboratory of Cryospheric Sciences, School of Architecture and Civil Engineering, École Polytechnique Fédérale de Lausanne, Lausanne, Switzerland.
[2]WSL Institute for Snow and Avalanche Research SLF, Davos, Switzerland.
[3]Environmental Remote Sensing Laboratory, School of Architecture and Civil Engineering, École Polytechnique Fédérale de Lausanne, Lausanne, Switzerland.
[4]Radar, Satellite, Nowcasting Departement, MeteoSwiss, Locarno, Switzerland.
[5]Institute of Meteorology and Climate Research, Atmospheric Environmental Research (KIT/IMK-IFU), KIT-Campus Alpin, Garmisch-Partenkirchen, Germany.
[6]unaffiliated, Sydney, Australia.

*Correspondence to:* F. Gerber (gerberf@slf.ch)

**Abstract.** Snow distribution in complex alpine terrain and its evolution in the future climate is important in a variety of applications including hydro-power, avalanche forecasting and fresh water resources. However, it is still challenging to quantitatively forecast precipitation especially over complex terrain, where the interaction between local wind and precipitation fields strongly affects snow distribution at the mountain ridge scale. Therefore, it is essential to retrieve high-resolution information about precipitation processes over complex terrain. Here, we present very high resolution Weather Research and Forecasting model (WRF) simulations (COSMO–WRF), which are initialized by 2.2 km resolution Consortium for Small-scale Modeling (COSMO) analysis. To assess the ability of COSMO–WRF to represent spatial snow precipitation patterns, they are validated against operational weather radar measurements. Estimated COSMO–WRF precipitation is generally higher than estimated radar precipitation, most likely due to an overestimation of orographic precipitation enhancement in the model. The high precipitation amounts also lead to a higher spatial variability in the model compared to radar estimates. Overall, an autocorrelation and scale analysis of radar and COSMO–WRF precipitation patterns at a horizontal grid spacing of 450 m show that COSMO–WRF captures the spatial variability normalized by the domain-wide variability of precipitation patterns down to the scale of few kilometers. However, simulated precipitation patterns systematically show a lower variability on the smallest scales of a few 100 m compared to radar estimates. A comparison of spatial variability for different model resolutions gives evidence for an improved representation of local precipitation processes at a horizontal resolution of 50 m compared to 450 m. Additionally, differences of precipitation between 2830 m above sea level and the ground indicate that near-surface processes are active in the model.

# 1 Introduction

In many regions of the world, e.g. the Alps or the Californian Sierra Nevada, snow is the main source of fresh water. Additionally, it is an important resource for hydro-power and is crucial for winter tourism in skiing areas (Schmucki et al., 2017). Thus, especially in a changing climate, it is essential to improve the understanding of processes forming the seasonal snow cover. Improving the ability of weather forecast models to represent the spatial variability of snowfall is further crucial to efficiently manage fresh water and hydro-power. Moreover, as snow is a potential danger in terms of avalanches, improved knowledge about the distribution of snow is crucial for avalanche forecasting and prevention.

Snow accumulation patterns at a mountain-range scale are known to be strongly dependent on blocking and lifting processes including large-scale orographic precipitation enhancement (e.g. Houze, 2012; Stoelinga et al., 2013), which is related to the large-scale atmospheric circulation. However, for a long time little knowledge was available about the spatial distribution of snow on a mountain-slope or river-catchment scale. Only in recent years improvements in technology allowed the investigation of mountain-slope scale snow distribution (e.g. Deems et al., 2006; Prokop, 2008; Grünewald et al., 2010). Terrestrial and airborne laser scanning reveal annually persistent patterns of peak-of-winter snow accumulation distribution on river-catchment scales (Schirmer et al., 2011; Scipión et al., 2013), which is found to be consistent with few dominant snowfall events of the season. Reported scale breaks in fractal analysis of snow accumulation patterns are mainly at scales < 100 m and represent the occurrence of a change in dominant processes (e.g. Deems et al., 2008). On very small-scales snow accumulation patterns are assigned to wind redistribution of snow (e.g. Mott et al., 2011; Vionnet et al., 2017). Vegetation effects were found to be dominant at small scales and terrain effects dominate on scales up to 1 km (Deems et al., 2006; Trujillo et al., 2012; Tedesche et al., 2017). Different dominant scales are reported for different slope exposions relative to the wind direction (Schirmer and Lehning, 2011). Furthermore, Schirmer et al. (2011) could show that snow accumulation smooths the underlying terrain, reducing the small-scale spatial variability of topography. While most studies addressed variability of snow accumulation, the combined scale analysis of snow accumulation and snow precipitation patterns by Scipión et al. (2013) reveals much smoother patterns in snow precipitation at about 300–600 m above ground compared to final snow accumulation at the ground on scales up to 2 km. This stresses the importance of pre-depositional near-surface and post-depositional processes for snow accumulation patterns.

Driving processes of snow accumulation on the mountain-ridge scale were addressed in numerous studies, which reveal two main pre-depositional processes. On the one hand, mountain-ridge scale precipitation and accumulation are influenced by local cloud-dynamical processes (Choularton and Perry, 1986; Dore et al., 1992; Zängl, 2008; Zängl et al., 2008; Mott et al., 2014). On the other hand, particle-flow interactions (i.e. the influence of the local flow field on the pathways of snow particles and particle distribution in the air) determine snow accumulation patterns in mountainous terrain (Colle, 2004; Zängl, 2008; Lehning et al., 2008; Dadic et al., 2010; Mott et al., 2010, 2014). On the mountain-ridge scale, Mott et al. (2014) documented the occurrence of a local event of orographic snowfall enhancement. In their case study, the presence of a low-level cloud gives evidence for precipitation enhancement favored by the seeder-feeder mechanism (e.g. Bergeron, 1965; Purdy et al., 2005). On similar scales, preferential deposition (Lehning et al., 2008) was found to cause enhanced snow accumulation on

leeward slopes (e.g. Mott et al., 2010; Mott and Lehning, 2010). However, snow depth measurements in very steep terrain and corresponding local flow field measurements reveal even more complex particle-flow interactions (Gerber et al., 2017) than previously suggested by model studies. On even smaller scales the main driver of snow accumulation patterns is post-depositional snow transport by drifting and blowing snow, which is dependent on local topographic features and wind gusts (Lehning and Fierz, 2008; Mott et al., 2010).

Complex terrain-flow-precipitation interactions (i.e. the effect of terrain-induced flow field variations on the precipitation formation and distribution), especially on the mountain-ridge scale, still leave the relative importance of the different pre-depositional processes for snow accumulation and the frequency of occurrence barely known (Mott et al., 2014; Vionnet et al., 2017). Running a coupled simulation of the snowpack model Crocus and the atmospheric model Meso-NH in large-eddy simulation (LES) mode, Vionnet et al. (2017) addressed the question of the relative importance of these different processes including snow redistribution by wind. Their results show that post-depositional snow transport dominates snow accumulation variability, but leaves the question of the relative importance of pre-depositional processes open.

Given the small scale of these processes their relative importance may either be addressed based on very high resolution numerical simulations or based on spatially highly resolved precipitation measurements. Therefore, accurate model results and radar measurements at high resolution are essential. Both, however, are challenging to achieve and very high resolution simulations are still rare especially over complex terrain. Remote sensing techniques, on the other hand, are the most important methods to obtain high-resolution spatial measurements of atmospheric properties at different atmospheric levels. They permit to gain information about both the small- and the large-scale properties of the atmospheric processes. The particular place among these techniques belongs to the weather radar, due to its wide coverage, fine spatial resolution, and interaction of microwaves with the precipitation. These properties have been used to infer orographic precipitation enhancement, particularly in the case of liquid precipitation (Panziera et al., 2015).

In this study, we present very high resolution WRF simulations, which are forced by 2.2 km resolution Consortium for Small-scale Modeling (COSMO) analysis and high resolution radar estimates making use of the recently renewed MeteoSwiss radar network (Germann et al., 2015) and its adequate technical performances, which allow observing precipitation in a challenging, complex alpine environment. Combining the COSMO–WRF simulations with operational radar measurements, we perform a variability analysis for snow precipitation at a regional to mountain-ridge scale to address the question: To what degree is snow precipitation variability represented by very high resolution WRF simulations?

Model simulations, radar measurements and analysis techniques are presented in Sect. 2. In a first part of the results and discussion (Sect. 3), we validate COSMO–WRF simulations against point measurements of temperature, relative humidity, wind speed and direction (Sect. 3.1). Spatial precipitation patterns in both, radar estimates and COSMO–WRF simulations, are presented in Sect. 3.2. Subsequently, we address the question to what degree the overall precipitation variability is represented in the model by analyzing the domain-wide statistics (Sects. 3.3). To address the spatial variability of precipitation patterns we present a discussion of dominant processes based on variograms and 2D-autocorrelation maps (Sect. 3.4). Variograms and autocorrelation analysis are widely used to address the spatial variability of snow accumulation and precipitation (e.g. Deems et al., 2008; Mott et al., 2011; Schirmer and Lehning, 2011; Scipión et al., 2013; Vionnet et al., 2017). While scale analysis

has been performed multiple times for snow accumulation patterns on a local scale, we address measured and modeled snow precipitation patterns at the approximate elevation of the operational weather radar on Weissfluhgipfel at 2830 m above sea level (m asl) on a mountain-ridge to regional scale. Additionally, we analyze modeled ground precipitation without taking into account any post-depositional processes. Given the different scales of analysis compared to previous studies, here we

address scales at which local cloud dynamics and particle-flow interactions are expected to occur but leave out scales at which snow accumulation is expected to be dominated by post-depositional snow redistribution. Following this analysis of spatial precipitation variability, which includes a discussion of dominant processes driving the spatial variability of precipitation patterns, Sect. 3.5 addresses the question if an increased model resolution may improve the representation of spatial variability in the model. Finally, our findings about the model performance, our analysis of the spatial variability of precipitation and

future perspectives are wrapped up in a conclusion and outlook (Sect. 4).

## 2  Data and Methods

### 2.1  WRF model setup

Atmospheric simulations are performed with the non-hydrostatic and fully compressible Weather Research and Forecasting (WRF) model (Skamarock et al., 2008) version 3.7.1 for the region of Eastern Switzerland (Fig. 1). Simulations are set up with

four one-way nested domains (d01–d04, Fig. 1). Domain d01 has a horizontal resolution of 1350 m, with 40 vertical levels and covers a region of a about 250 km times 320 km including eastern Switzerland and a portion of the neighboring countries (Fig. 1, Table 1, Supplementary Information S1). The three nests have horizontal resolutions of 450 m, 150 m and 50 m using a nesting ratio ($dx_{parent}/dx_{nest}$) of 3. Domains d02–d04 have 40, 60 and 90 vertical levels, respectively, with the model top at 150 hPa using a preliminary version of vertical nesting (Daniels et al., 2016). Twenty and 40 vertical levels are refining

the whole atmosphere in domains d03 and d04, respectively. Ten vertical levels in d04 are introduced to additionally refine the boundary layer. To make sure that there is plenty of domain for the model to adapt to the refined topography, domain d02 is shifted toward the eastern boundary of domain d01 as dominant wind directions are from a north-westerly and southerly direction. Domain d02 covers the central northern part of the Grisons, while domains d03 and d04 cover the surroundings of Davos and the upper Dischma valley, respectively (Fig. 1). Simulations are performed for three snow precipitation events on

31 January 2016, 4 February 2016 and 5 March 2016 (Sect. 2.5).

The parent domain is run with a planetary boundary layer (PBL) scheme (non-large eddy simulation (non-LES) mode), while the three nests are run in the LES mode. No strong differences were found when running domains d02 and d03 in non-LES mode (not shown). Therefore and as we are interested in having an as good as possible representation of small-scale winds, we decided to run our simulations in the LES mode for all nested domains. Domains d02 and d03 are within the 'gray zone' (Wyngaard,

2004). There are approaches omitting simulations in the 'gray zone' by the choice of a higher grid refinement ratio (Muñoz Esparza et al., 2017), which would be worth a sensitivity study. However, we use the well-tested 1:3 grid refinement ratio and keep our model setup consistent with the very high resolution simulations by Talbot et al. (2012), except that they perform separate simulations for the non-LES and LES domains, while we run a nested simulation with one-way feedback for all four

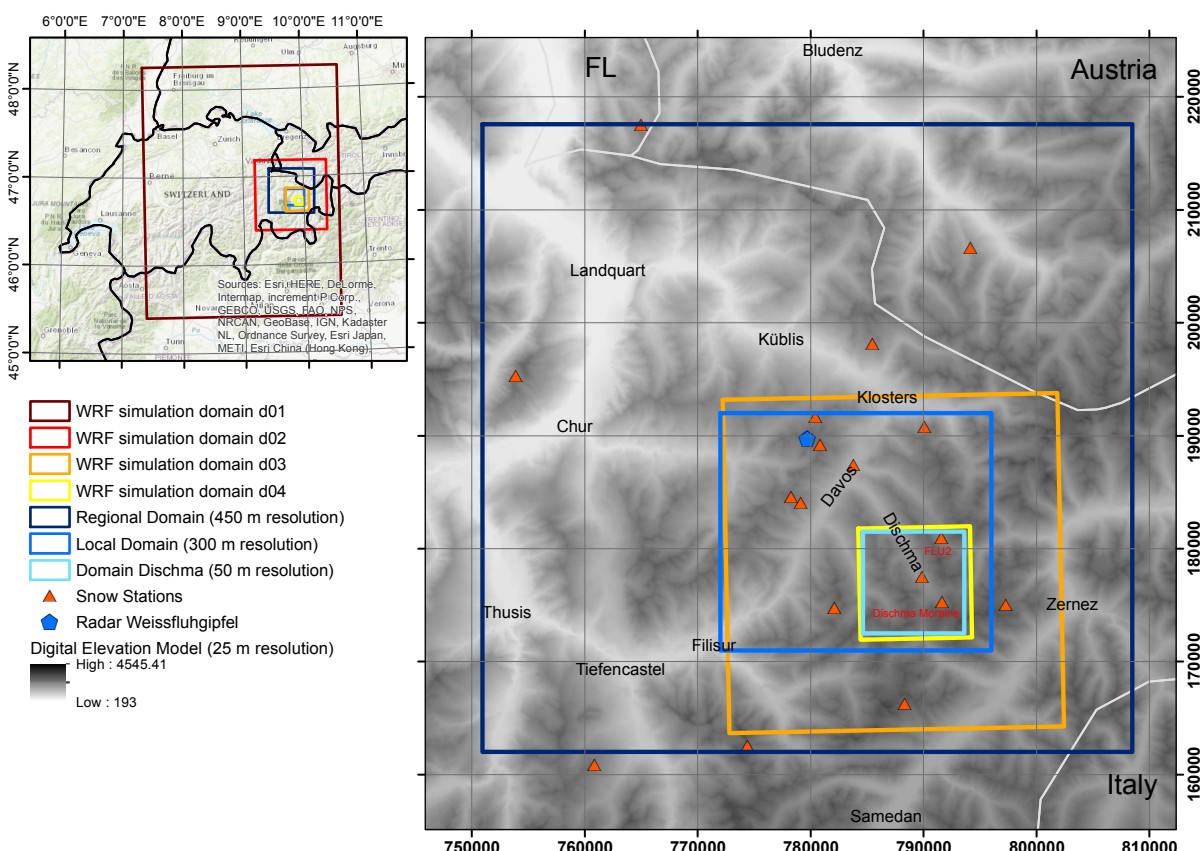

**Figure 1.** Overview over the study area in the eastern part of Switzerland surrounding Davos. WRF simulation domains (d01–d04, dark red to yellow) and evaluation domains (blue) give information on the simulation and evaluation setup. The 18 meteorological stations (red triangles) are within or very close to the regional domain. The two stations *Dischma Moraine* and *FLU2*, which are used to validate the model, are within domain Dischma. The operational weather radar is located on Weissfluhgipfel at approximately 2830 m above sea level (m asl, blue pentagon). Coordinates in the right panel are in Swiss coordinates CH1903LV03 (unit: m). Shaded topography: dhm25 © 2018 swisstopo (5740 000 000).

domains. Running a nested simulation of the non-LES and LES domains turned out to be necessary for precipitation to evolve properly in the LES domains, as hydrometeors cannot be used as a boundary condition for the parent domain but are fed to nested domains in WRF simulations. Subgrid-scale turbulence is parametrized by the 1.5 order turbulent kinetic energy closure (Skamarock et al., 2008). For the non-LES setup the Yonsei University PBL parameterization (YSU PBL, Hong et al., 2006) is used, which is considered to be one of the schemes showing the best performance over complex terrain (Gómez-Navarro et al., 2015). An adapted version of YSU PBL was shown to perform even better when taking into account subgrid-scale variability of the terrain (Jiménez and Dudhia, 2012; Gómez-Navarro et al., 2015). However, given our high model resolution we decided to keep the model simple and run the simulations with the standard YSU PBL. Landuse data is taken from the Corine dataset

(European Environmental Agency, 2006) and translated to the USGS conventions (Pineda et al., 2004; Arnold et al., 2010). Soil type is set to *silty clay loam* for the whole domain. The link between the soil, which is modeled by the Noah land-surface model with multi-parameterization options (Noah-MP, Niu et al., 2011; Yang et al., 2011), and the atmosphere is given by the MM5 Monin-Obukhov surface layer model (Paulson, 1970; Dyer and Hicks, 1970; Webb, 1970; Zhang and Anthes, 1982; Beljaars,

1994), which is based on the Monin-Obukhov similarity theory (Obukhov, 1971). For microphysics the Morrison 2-moment precipitation scheme (Morrison et al., 2005, 2009) is used, which was found to be one of the schemes, which most adequately simulate snow precipitation over complex terrain (Liu et al., 2011). Details about processes in the Morrison parameterization are given in Appendix A. An investigation of different microphysical parameterizations would be interesting, but is beyond the scope of this study. Given the high horizontal resolution no sub-grid parameterizations for cumulus clouds is used.

The 2.2 km horizontally resolved Consortium for Small-scale Modeling (COSMO–2) analysis by MeteoSwiss are used as initial and boundary conditions for the parent domain. For COSMO–2 analysis data to be readable by the WRF pre-processing system a regridding of the rotated COSMO-coordinates to latitude-longitude coordinates is required. COSMO preprocessing, model adaptations and details about the model simulations are given in Gerber and Sharma (2018).

Topography in the model is based on the Aster Global Digital Elevation Model V002 with a resolution of one arc-second

(METI/NASA, 2009). Terrain smoothing has been applied for all domains due to the very steep terrain in the simulation area. Four cycles of the WRF 1–2–1 smoothing (i.e. a moving window filter with a window length of 3 and weights of 1:2:1 for the grid points i-1, i and i+1) are applied to all four domains to keep all slopes in domain d04 (50 m horizontal grid spacing) below 45°. Additionally, the boundaries of the parent domain are smoothed to match COSMO-topography (Gerber and Sharma, 2018). Test simulations are run with 14 cycles of WRF 1–2–1 smoothing, which allows for a longer computational timestep

and therefore saves computational time (Table 1). Maximum slope angles for all domains and different smoothing are given in Table 1. Simulations with different precision of topography further allow us to address the importance of the representation of topography in the model. To allow the simulations to adapt to higher resolution topography domains d01–d04 are run with a spin-up of 43 h, 19 h, 7 h and 1 h, respectively.

As the snow cover in complex alpine terrain is likely rougher than for a flat field and to account for non-resolved topography

and additional smoothing, snow surface roughness length has been changed to 0.2 m. The chosen roughness length is much larger than roughness lengths assumed by e.g. Mott et al. (2015). However, grid spacing in our simulations is larger and the roughness length is chosen such that it accounts for roughness elements in complex terrain (e.g. large rocks) and non-resolved topography, which are assumed to have an average size of about 2 m. This estimate is based on a comparison of a 2-m digital terrain model (DTM-AV © 2018 swisstopo (5704 000 000)) to a 25-m resolution digital elevation model (dhm25 © 2018

swisstopo (5740 000 000)), which reveals an average difference on the order of 2.5 m for bare ground conditions in domain d04 between 2200 and 2700 m asl. Hence, the estimate of 2 m is rather conservative but takes into account smoothing of the terrain by the snow cover.

For the model validation (Sect. 3.1) WRF variables, using model output of the four grid points surrounding the station, are linearly interpolated to the coordinates of the meteorological station (see Sect. 2.2). Alternatively, the eight neighbors of the grid

point closest to the station could be used (Goger et al., 2018). Temperature is corrected for elevation due to terrain smoothing

**Table 1.** Setup for the four nested domains (d01–d04) used in the WRF simulations. For the planetary boundary layer (PBL) the simulation mode is given, distinguishing between non-large eddy simulation (LES) and LES settings. For non-LES settings the PBL scheme is given. Additionally, the subgrid scale (SGS) turbulence parameterization is given for all four domains. $dx$, $dy$ give the horizontal resolution. *Vertical levels* gives the number of vertical levels in the different domains. *Vertical levels (< 1000 m)* gives the number of vertical levels in the lowest 1000 m of the atmosphere. The lowest 21 model levels for all four domains are given in Table S.1. The time step ($dt$) and the *maximum slope angle* are given for simulations with 4 (14) smoothing cycles.

| Domain | PBL mode | PBL scheme | SGS scheme | dx, dy (m) | Vertical levels | Vertical levels (< 1000 m) | dt (s) | Max. slope angle ° |
|--------|----------|------------|------------|------------|-----------------|----------------------------|--------|--------------------|
| d01 | non-LES | YSU[1] | 1.5 order TKE[2] closure | 1350 | 40 | 8 | 1 (6) | 17.5 (9.9) |
| d02 | LES | - | 1.5 order TKE[2] closure | 450 | 40 | 8 | 1/3 (2) | 35.2 (26.5) |
| d03 | LES | - | 1.5 order TKE[2] closure | 150 | 60 | 9 | 1/9 (1/2) | 39.8 (36.8) |
| d04 | LES | - | 1.5 order TKE[2] closure | 50 | 90 | 21 | 1/27 (1/4) | 44.5 (37.4) |

[1]YSU: Yonsei University PBL scheme, [2]TKE: Turbulent kinetic energy

using a moist-adiabatic temperature gradient of $-0.0065 \text{Km}^{-1}$. Modeled wind speeds are extrapolated to the measurement height by applying a logarithmic wind profile, as wind measurements at the automatic weather stations are not taken at 10 m but 4 or 5 m above ground (Sect. 2.2). This is a rough approximation given the assumption of a neutral atmosphere. For simulation domains d01–d03 10-m wind speeds are extrapolated to the elevation of the sensor above the snow cover, while for domain d04 wind speeds at the lowest model level (approximately 3 m above ground) are used for the extrapolation. The dynamic reference roughness length is chosen to be 0.2 m (corresponding to the surface roughness length in the model simulations). For wind direction comparisons wind directions at 10 m and 3 m above ground are chosen for the simulation domains d01–d03 and d04, respectively. As a reference COSMO–2 variables of the closest grid point to the station are included in the model validation and hence in Fig. 2 and Fig. 3. Two-meter temperature and 10-m wind speed of COSMO–2 are corrected for elevation by the same procedure as for the WRF simulations.

## 2.2 Automatic weather stations

Snow depth measurements from a total of 18 automatic weather stations in the central northern part of the Grisons (Fig. 1) are used. Two stations (*Dischma Moraine* and *Dischma Dürrboden*) were installed as part of the Dischma Experiment (DISCHMEX), in which processes of snow accumulation and ablation in the Dischma valley near Davos (Switzerland) are addressed (Gerber et al., 2017; Mott et al., 2017; Schlögl et al., 2018). 16 stations are part of the Intercantonal Measurement and Information System (IMIS). The 18 stations are located between 1560 m asl and 2725 m asl. The stations measure snow depth in addition to the standard meteorological parameters. All stations have shielded temperature and humidity sensors, but are unheated. Biased temperatures around midday and occasional data gaps due to iced instruments may therefore occur (Huwald et al., 2009; Grünewald et al., 2012). Two stations (*Dischma Moraine* and *FLU2*), which are located in the WRF domain d04

with a horizontal grid spacing of 50 m, are used for the model validation. The variables evaluated are 2-m temperature, 2-m relative humidity, wind speed and wind direction. Wind measurements at IMIS stations are taken about 5 m above ground, while the wind sensor at station *Dischma Moraine* is located at about 4 m above ground.

## 2.3 Operational weather radar data

Weather radar datasets employed in the presented analyses are acquired by the MeteoSwiss operational radar located at the Weissfluhgipfel (2850 m asl), in the proximity of Davos. It is a dual-polarization Doppler weather radar, providing complementary information about the detected hydrometeors by considering their interaction with the incident electromagnetic radiation in both, horizontal and vertical, polarization planes. This complementary information leads to an enhanced clutter detection, which makes the radar measurements in such a complex mountainous terrain significantly more reliable. The polarimetry also
makes it possible to identify the type of hydrometeors (Besic et al., 2016), which allows us to be confident that in the zone of interest for the presented study we deal with solid precipitation, consisting mostly of aggregates and crystals, and partly of rimed ice particles.

   The radar operates in 5-minutes cycles during which it scans the surrounding atmosphere by performing complete rotations at twenty different elevations, from $-0.2°$ to $40°$ (Germann et al., 2015). Operationally, the size of a radar sampling volume
is $500$ m in range, whereas the size observed in the perpendicular plane depends on the half-power beamwidth and increases with range. The acquired data undergo an elaborated procedure of corrections (Gabella et al., 2017). Before the quantity of precipitation at the ground level is estimated by averaging over $1$ km$^2$ the observations are corrected for the Vertical Profile of Reflectivity (VPR) with the weight assigned to volumes being inversely proportional to their height above the ground (Germann et al., 2006).

In the framework of our study, rather than relying on the operational radar product, we use data with the highest available resolution of $83$ m in range. We also adopted a more conservative, non-operational method of clutter identification, which relies exclusively on the polarimetry and leaves very little residual clutter, however, sometimes at the expense of removing some precipitation. Given that we consider only radar measurements at low elevation angles in the vicinity of the radar and that the bright band is not present in our case studies (all radar measurements are from above 2800 m asl during the winter season),
the observations are not corrected for the VPR. Furthermore, given the strong influence of wind on the snow precipitation, we restrict our precipitation estimate to only four elevations, from the second to fifth ($0.4°, 1°, 1.6°, 2.5°$), avoiding the first one, judged to contain too little information due to the abundant rejected ground clutter areas.

   Polarimetry helps to identify non-meteorological scatterers, to distinguish between different types of hydrometeors, to correct for signal attenuation and to make quantitative estimates of intense to heavy rainfall. For snowfall measurements it is
common to use reflectivity $Z$ at horizontal polarization and convert it into snow water equivalent $S$ using a so-called Z-S relationship (Saltikoff et al., 2015):

$$Z = 100S^2. \tag{1}$$

The coefficients used in this formula account for the dielectric properties and fall velocities of snow and convert reflectivity $Z$ in snow water equivalent $S$. The radar provides an indirect estimate of snowfall, rather than a direct measurement. Applied on each radar sampling volume scanned by the four selected elevations in the zone of interest (up to 40 km around the radar), the formula gives an estimate of liquid precipitation equivalent in the three-dimensional volume. By vertically averaging estimates from the four elevation sweeps using equal weights, we obtain the estimate of precipitation in polar (range, azimuth) coordinates at a flat plane at the height level of the radar. These estimates are summed up over 24 h to get the accumulation maps used in the study.

Further on, the polar accumulation maps are re-sampled by means of the bi-linear interpolation to the Cartesian grid of the regional domain (450 m resolution) and the local domain (300 m resolution). The obtained Cartesian maps are finally processed to remove the residual clutter using a 3×3 median filter, partly or entirely. The former means that only the isolated high values in the original map are replaced with the corresponding value of the filtered map, at the positions where the difference between the original and the filtered map appears to be larger than 5 mm (hereafter "partly-filtered"). The latter means that the entire map is influenced by the median filtering (hereafter denoted as "entirely-filtered").

## 2.4 Autocorrelation and variogram analysis

To investigate the variability of snow precipitation and accumulation patterns and their relation to topography a scale analysis, based on 2-dimensional (2D) autocorrelation maps and variograms, is performed. 2D-autocorrelation maps and variograms are further used to relate variability in radar and WRF precipitation. Given the resolution restriction by the radar measurements (Sect. 2.3) we analyze two different domains using horizontal resolutions of 450 m and 300 m, respectively. The domain with a resolution of 450 m covers an area of about 58 km times 56 km centered over the radar on Weissfluhgipfel (hereafter regional domain, Fig. 1). The domain with a resolution of 300 m covers an area of 24 km times 21 km to the south of Davos (Switzerland) including the Dischma valley (hereafter local domain). Radar data (300 m resolution) and WRF precipitation on three resolutions (450 m, 150 m and 50 m) are additionally, evaluated on domain Dischma to address the influence on the spatial resolution of variability. Domain Dischma covers the upper Dischma valley with an extent of 9 km times 9 km.

To produce variograms the semivariance ($\hat{\gamma}$) is calculated at 50 logarithmic lag distance bins ($h$, i.e. a set of distance ranges) by

$$\hat{\gamma}(h) = \frac{1}{2|N(h)|} \sum_{(i,j) \in S(h)} (a_j - a_i)^2, \qquad (2)$$

where $S(h)$ are the point pairs $(i,j)$ and $N(h)$ gives the number of point pairs of the evaluated variable $a$. WRF and radar snow precipitation and topography are evaluated at 450 m and 300 m resolutions with a maximum lag distances of 25 km and 10 km, respectively. Variograms for domain Dischma are calculated with a maximum lag distance of 5 km. Minimum numbers of point pairs in one lag distance bin for the local and regional domain are 18317 and 8035, respectively. For domain Dischma the minimum number of point pairs is between 677 and 55419, depending on the resolution.

To determine scaling properties an empirical log-linear model is fit to the variogram by least square optimization (Schirmer et al., 2011). The model used is not a valid variogram model but used to describe the experimental variograms and chosen to

be consistent with Schirmer et al. (e.g. 2011). For all variograms three empirical log-linear models are fit:

$$y(x) = \begin{cases} \alpha_1 * \log(h) + \beta_1, & \text{for } \log(h) < l_1 \\ \alpha_2 * \log(h) + \beta_2, & \text{for } l_1 \geq \log(h) < l_2 \\ \alpha_3 * \log(h) + \beta_3, & \text{for } \log(h) \geq l_2 \end{cases} \tag{3}$$

using the constraint that each log-linear model needs to contain a minimum of four data points and the continuity constraint(s)

$$\alpha_1 \log(l1) + \beta_1 = \alpha_2 \log(l1) + \beta_2$$
$$\alpha_2 \log(l2) + \beta_2 = \alpha_3 \log(l2) + \beta_3, \tag{4}$$

where $\alpha_{1,2,3}$ and $\beta_{1,2,3}$ are the slopes and intercepts of the three log-linear models, respectively. Scale breaks ($l_1$, $l_2$) are the lag distances of the intersections of the first and second, and second and third log-linear model, respectively. Scale breaks were previously found to determine the scale of a change of dominant processes (e.g. Deems et al., 2006). To address the variability with respect to the overall variability in the respective domain, all variograms are normalized by the total domain-wide variance.

2D-autocorrelation is calculated based on Pearson's correlation coefficient $r$ of all grid point pairs for a maximum lag distance of $\pm40$ grid points in x- and y-direction. This results in maximum lag-distances of 18 km for the regional domain.

## 2.5 Snowfall events

This study is based on three precipitation events in winter 2016. On 31 January 2016 the Azores high and a low-pressure area over Scandinavia induce westerly flow over central Europe and relatively mild temperatures with about $-3°C$ at 2500 m asl. A shift of the Azores high toward northern Spain and a trough over eastern Europe lead to a change in wind direction toward northerly advection and a decrease of temperature (about $-12°C$ at 2500 m asl) on 4 February 2016. On 5 March 2016 a low-pressure area over France, which is part of a large depression area over central Europe, causes southerly advection over Switzerland. Temperatures are about $-7°C$ at 2500 m asl. Given the relatively high temperatures on 31 January 2016, which resulted in quite substantial liquid precipitation at the lowest elevations, total (solid and liquid) ground precipitation is evaluated. This does not make a big difference for the precipitation events on 4 February 2016 and 5 March 2016 but is essential for the precipitation event on 31 January 2016. For precipitation patterns at the elevation of the radar (2830 m asl) we only analyze solid precipitation from WRF.

## 3 Results and Discussion

## 3.1 Point validation of WRF simulations

Two-meter air temperature and relative humidity, and 4- or 5-m wind speed and direction at two stations (Sect. 2.2) are compared to WRF to validate the model (Fig. 2 and Fig. 3). For both stations 2-m temperature matches reasonably with observations although especially for the precipitation event on 4 February 2016 substantial temperature deviations occur around

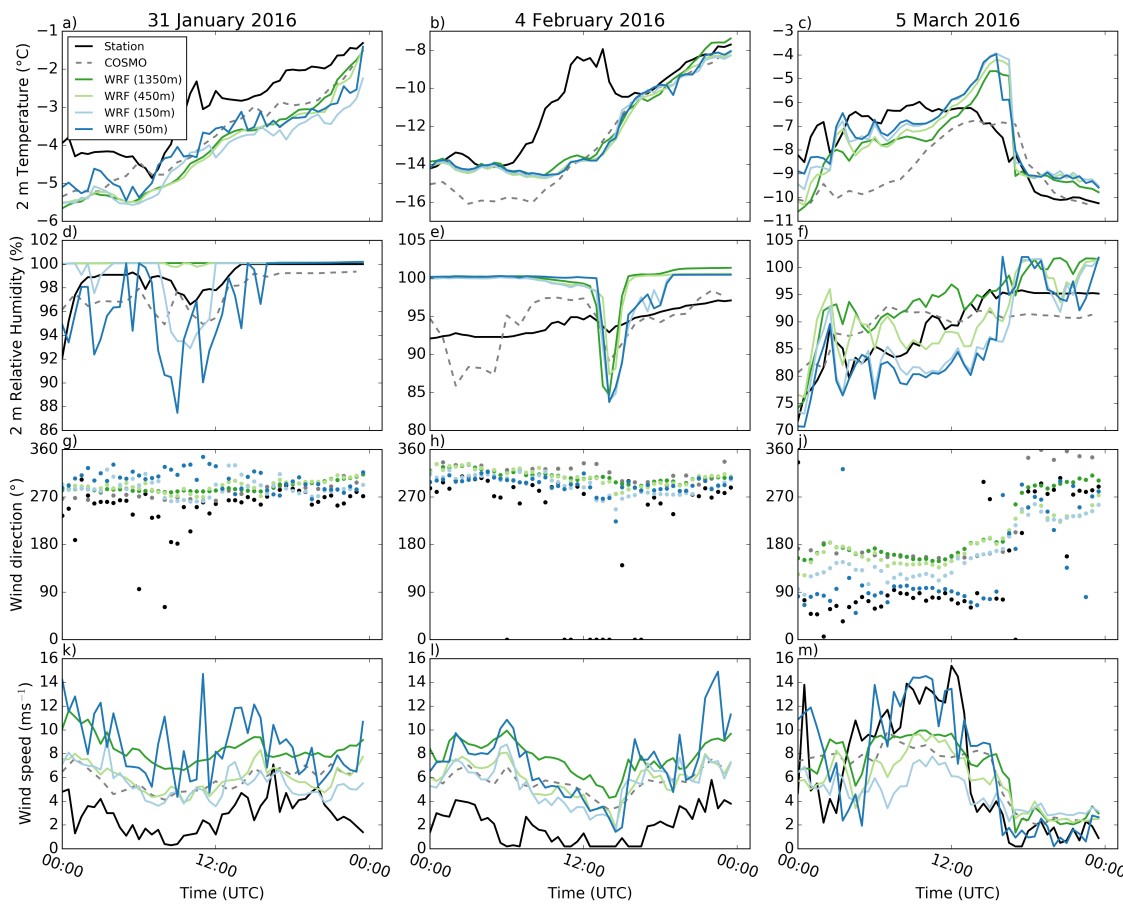

**Figure 2.** Comparison of 2-m temperature (C$^\circ$), 2-m relative humidity (%), 4-m wind speed (ms$^{-1}$) and wind direction ($^\circ$) at station *Dischma Moraine* (black) to WRF simulations interpolated to the coordinates of station *Dischma Moraine* for the three precipitation events on 31 January 2016, 4 February 2016 and 5 March 2016 for all four simulation domains (d01: dark green, d02: light green, d03: light blue, d04: dark blue). For comparison COSMO–2 is added for the closest grid point (dashed gray). Two-meter temperature in WRF and COSMO are corrected for elevation based on a moist-adiabatic temperature gradient.

midday (Fig. 2a-c and Fig. 3a-c). Deviations of the WRF model from station measurements during midday are likely caused by errors in station measurements due to radiative heating of the multiplate shielded temperature sensors (Huwald et al., 2009, Sect. 2.2).

Relative humidity shows partially good agreement, but shows a strong temporal variability (Fig. 2d-f and Fig. 3d-f). WRF is
5 generally able to capture main drops in relative humidity at the two investigated stations but it introduces additional drops com-

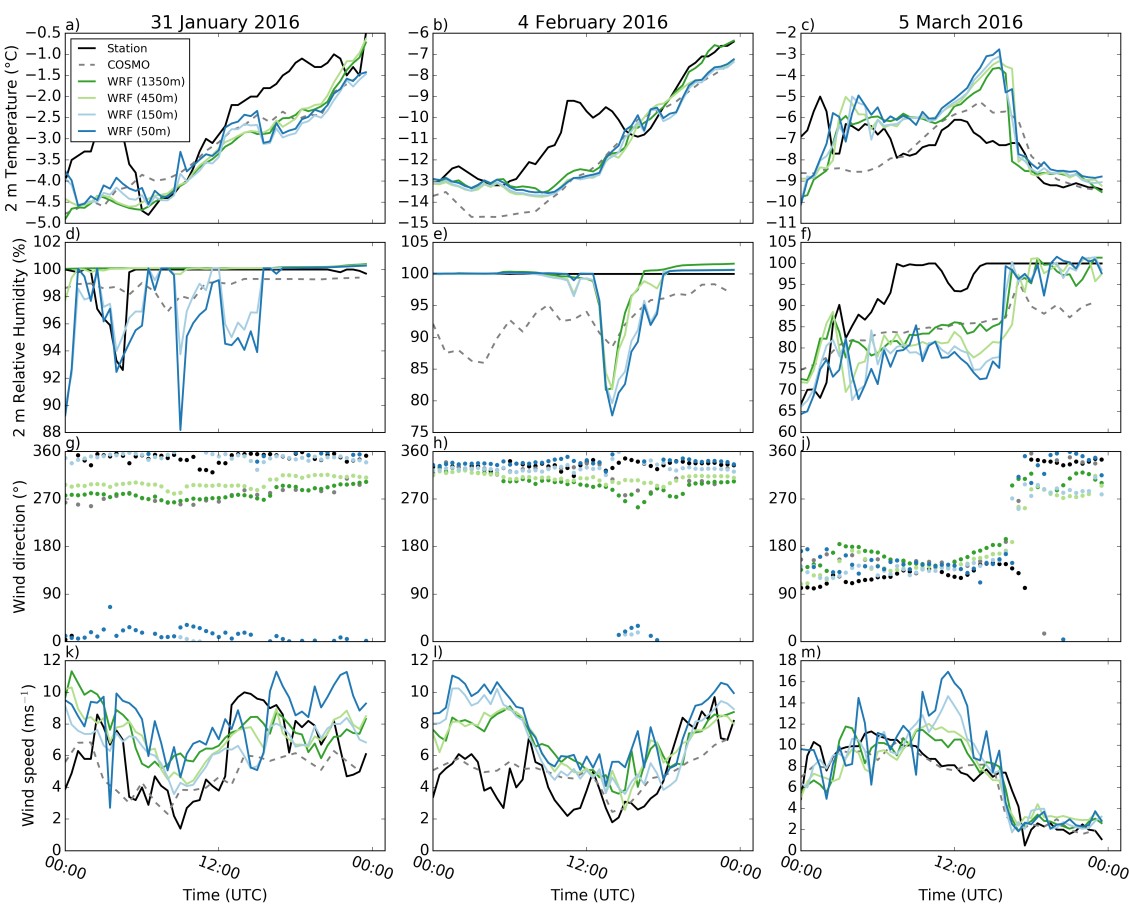

**Figure 3.** As Fig. 2 but for the station *FLU2* on the Flüelapass with 5-m wind speed and direction.

pared to measurements. The microphysics parameterization is originally developed for simulations with a coarser resolution, which produce less vertical motions. Thus, the introduction of a higher variability in relative humidity in our WRF simulations may be due to strong subsidence and lifting, which lead to an overestimation of adiabatic cooling or warming and hence to an overestimation of humidity generation or decay. Additionally, differences between modeled and measured relative humidity may be due to measurement uncertainties.

5   Simulated wind direction shows good agreement with measured wind direction (Fig. 2g-i and Fig. 3g-i). In complex terrain, simulations are often limited to resolutions, which are too coarse to resolve smaller-scale terrain features that affect near-surface wind direction (e.g. due to a lack of high resolution terrain data, or computational resources), and thus cannot accurately capture changes in wind direction close to the surface where weather stations are located. Good agreement in wind direction modeling

in our COSMO–WRF simulations in complex terrain is likely due to the high resolution of topography. For some cases wind directions in the WRF simulations additionally improve for higher resolutions although for others terrain smoothing is likely to have adverse effects on modeled wind direction.

Compared to the good agreement of wind direction, wind speeds show only partially good agreement with station measurements (Fig. 2k-m and Fig. 3k-m). Wind speeds were found to strongly depend on the subgrid-scale turbulence parameterization and a strong overestimation of wind speeds was observed for different simulation setups (not shown). Applying the improved non-linear subgrid-scale turbulence parameterizations (Mirocha et al., 2010, 2014) leads to instabilities in the current model setup. The use of a snow surface roughness length of 0.2 m, representing the combined roughness of snow and surface features (e.g. boulders and rocky outcroppings, Sect. 2.1), compared to simulations with standard WRF roughness length of snow of 0.002 m, could partially reduce overestimated wind speeds (Gerber and Sharma, 2018). While we address non-resolved topography based on an increased snow surface roughness length, another approach to improve wind speeds in WRF simulations has been introduced by Jiménez and Dudhia (2012), who use a sink term in the momentum equation based on subgrid-scale topography. They demonstrate the ability of their approach to improve surface wind speeds. However, with this approach, the effect of the subgrid scale topography is only included for simulations using a PBL parameterization. As in our model setup a PBL parameterization is only applied for domain d01, we address the non-resolved topography by increasing the surface roughness, which allows us to include the effect of non-resolved topography for all four simulation domains. In addition, our simulations are run over snow covered terrain, which implies that the standard roughness length for snow used in WRF (on the order of $10^{-3}$ m) is two orders of magnitude lower than roughness lengths representative of the scale of complex terrain in our simulations (on the order of $10^{-1}$ m). Applying the PBL version of Jiménez and Dudhia (2012) might be a possibility to reduce excess wind speeds in domain d01, which might also impact wind speeds in the domains d02–d04. However, such a sensitivity study is out of scope of the present investigation.

Based on our approach COSMO–WRF still simulates excess wind speeds for the two precipitation events on 31 January 2016 and 4 February 2016 and for the precipitation event on 5 March 2016 at station FLU2 for a few hours around noon. The overestimation on 31 January 2016 and 4 February 2016 is assumed to be connected to the upwind location of both stations during these two precipitation events, as speed up over windward slopes and ridges are a known problem (Mott et al., 2010; Gómez-Navarro et al., 2015). Hence, the exact location of the station relative to the ridge is important to verify wind speeds. Furthermore, local terrain features upstream of the station may disturb the wind field. For example station Dischma Moraine is located on a moraine on the northern side of the ridge between Piz Grialetsch and Scalettahorn. Station FLU2 is located on the northern side of Flüelapass above a small rock face and to the east of a terrain knoll. Such terrain features, while not represented in the model, may strongly reduce wind speeds in reality. Another potential cause of disagreement is that station measurements are prone to measurement uncertainties and riming of the unheated instruments may lead to an underestimation of wind speeds (Grünewald et al., 2012).

Generally, reasons for an overestimation of wind speeds may be manifold. An exact estimation of wind speeds at stations in the model is not expected due to unresolved topographical features in the complex terrain of our study site. An additional source of uncertainty – though unlikely to be on the order of the strong excess wind speeds – is the extrapolation of wind speeds

based on the assumption of a neutral atmosphere. While different potential causes of wind speed overestimation are discussed above, actual reasons for deviations in wind speed remain unknown.

While the model is designed such that it develops independently (given its fetch distances and spin-up times, Sect. 2.1), a poor representation of the large-scale gradients in the COSMO–2 input might not be corrected by COSMO–WRF. The inves-
tigated variables do not show a consistent signal of improvement nor a consistent signal of worsening with higher resolutions and with respect to the COSMO–2 input. Similarly but not consistently in phase with COSMO–WRF, COSMO–2 shows a good agreement with station measurements for certain cases, while it shows worse performance for other cases. Given these inconsistencies between cases and for COSMO–2 input a poor representation of the large-scale gradient in COSMO–2 is an unlikely reason for a bad model performance.

Overall, we show that the presented simulation setup captures temperature, relative humidity and wind conditions in complex terrain at two stations by a certain degree. Temperature deviations around midday are likely due to measurement uncertainties. Wind speeds tend to be overestimated, especially on the windward side of the mountain ridges. This shows that even for very high model resolutions the point-performance of the model with respect to wind speeds remains challenging. However, the model shows a good performance in the representation of the local wind directions, which is likely a consequence of the improved representation of topography.

### 3.2 Spatial snow precipitation and accumulation patterns

Radar precipitation maps of the regional domain covering an area of about 58 km times 56 km centered over the radar on Weissfluhgipfel (Fig. 1) tend to show wind direction (Fig. 2g-i, Fig. 3g-i and Fig. 4d-f) dependent precipitation patterns (Fig. 4). The precipitation field on 31 January 2016 shows a strong south-north gradient (Fig. 4a), while the precipitation field on 4 February 2016 shows a more homogeneous distribution (Fig. 4b). For the precipitation event on 5 March 2016 radar precipitation maxima are observed over the mountain ridges in the southern part of the domain (Fig. 4c). Although our regional domain does not represent a cross section across the whole alpine mountain range, a north-south (south-north) precipitation gradient for southerly (northerly) advection are apparent. This is in good agreement with large-scale orographic precipitation enhancement (Houze, 2012; Stoelinga et al., 2013), which favors precipitation on the upwind side of a mountain range due to topographically induced lifting and a drying due to sinking air masses downwind of the mountain range.

These large-scale patterns of orographic precipitation enhancement are partially captured in the WRF simulations (Fig. 4d-f). Especially, for southerly advection (precipitation event on 5 March 2016) this large-scale effect is well represented in COSMO–WRF, where precipitation maxima occur over mountain ridges in the southern part of the domain and a north-south precipitation gradient is present. For northerly to north-westerly advection (precipitation events on 31 January 2016 and 4 February 2016), however, snow precipitation maxima in the WRF simulations are shifted eastward compared to radar precipitation estimates, i.e. toward the outflow boundary.

Microphysics and precipitation dynamics in the model are likely to be a limiting factor in terms of small-scale precipitation patterns. Disagreement between radar and WRF precipitation patterns may further be connected to the strong terrain smoothing in the model. Despite of the high resolution of our simulations, slope angles are relatively low with maximum slope angles of

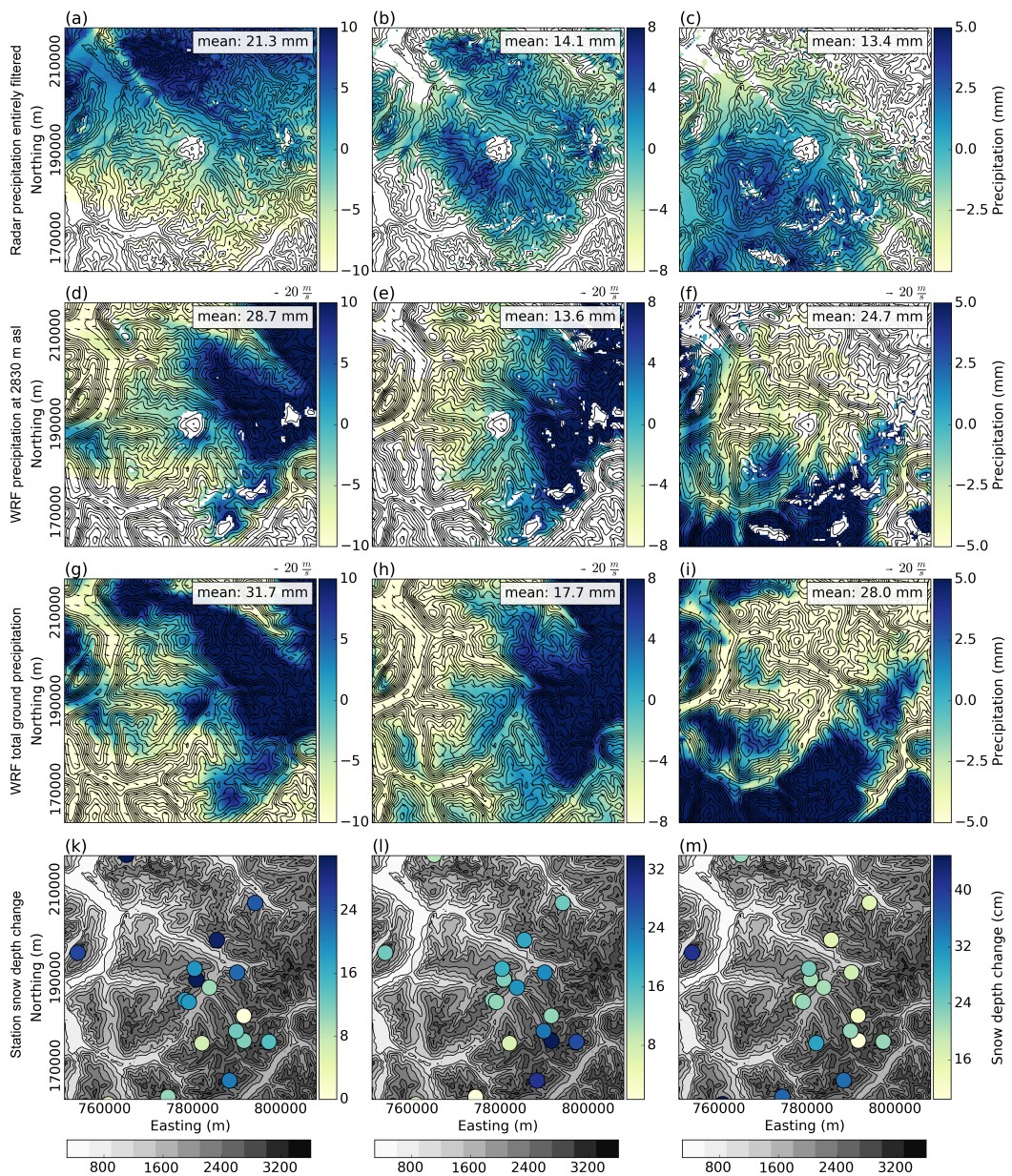

**Figure 4.** Twenty-four hour snow precipitation (mm) from a-c) MeteoSwiss entirely-filtered radar measurements, d-f) Weather Research and Forecasting (WRF) snow precipitation at 2830 m above sea level (m asl), g-i) WRF total ground precipitation and k-m) 24 h snow depth changes (cm) at meteorological stations on 31 January 2016 (left), 4 February 2016 (middle) and 5 March 2016 (right) with a resolution of 450 m in the regional domain (Fig. 1). Radar precipitation is estimated from different radar elevations (Sect. 2.3). White areas in a-f mark areas where clutter is removed and small values in the radar data are masked. The same mask is applied for WRF solid precipitation at 2830 m asl (approximate elevation of the radar, d-f), for which additionally areas where WRF topography is higher than 2830 m asl are masked. Arrows in d-f indicate wind direction and speed at an elevation of 2830 m asl. Northing and easting are given in the swiss coordinate system (CH1903LV03). Note different colorbars. Contour lines in a-c) and k-m): dhm25 (c) 2018 swisstopo (5740 000 000). Gray shading in k-m) represent topography.

35.2° in the regional domain due to the application of terrain smoothing (Table 1). Given even lower slope angles in domain d01 precipitation fed to domain d02 may already be too weak and thus needs to develop within domain d02. As mountains in the north-western part of the domain are shallower than mountains in the south-eastern area (Fig. 1), lifting condensation may not be strong enough in the north-western area of the domain, leading to precipitation generation further downstream in the domain, where steeper and higher mountains may even lead to too strong precipitation enhancement. Additionally, if the tendency of overestimated wind speeds sustains up to higher atmospheric levels in the model, this may lead to an overestimation of the advection of hydrometeors in the microphysics scheme (Morrison et al., 2005). This would result in a downstream shift of the precipitation maximum. However, we do not expect this to have a strong impact on the regional scale precipitation distribution. Thus, there are likely additional reasons for the observed downstream shift of WRF precipitation compared to radar precipitation, which remain difficult to explain.

On a mountain-valley scale (local domain) the same tendencies emerge with good agreement in overall gradients for southerly advection and partially reversed gradients for northerly to north-westerly advection when comparing WRF to radar precipitation patterns (not shown). WRF precipitation patterns generally show a stronger dependency on topography expressed in higher precipitation rates over higher elevations. Radar precipitation patterns additionally reveal small-scale precipitation patterns. Very small-scale patterns are visible on the partly-filtered radar maps (Sect. 2.3, not shown), while in entirely-filtered radar estimates (Fig. 4a-c) smallest-scale patterns are eliminated but patterns of about 1 km size emerge. Patterns in the entirely-filtered data could be small-scale precipitation cells, while the very small-scale patters are most likely noise in the radar data (see Sect. 3.4).

New snow depth measured at 18 automatic weather stations in the regional domain (Fig. 4k-m) over 24 hours shows a distinct elevation gradient, which is quite well represented by WRF total ground precipitation (Fig. 4g-i). For 31 January 2016 and 5 March 2016 the large-scale precipitation trend observed in the radar data is generally represented in station measurements. On 4 February 2016 station measurements suggest a precipitation peak in the upper Dischma valley (lower left quadrant in Fig. 4l), which agrees with WRF simulations. Radar estimates, however, show a more homogeneous distribution of precipitation on 4 February 2016. Snow depth changes at the stations are very local and strongly affected by wind redistribution of snow, which may disturb the large-scale gradient. Additionally, the distribution of stations is not homogeneous over the regional domain and fewer stations are available in the western part of the domain.

The visual comparison of radar and WRF precipitation patterns for all three events (Fig. 4) reveals that precipitation patterns are influenced by wind direction and topography. Large-scale precipitation patterns are in agreement with station measurements, although the latter are strongly influenced by the local wind field and snow redistribution processes.

## 3.3 Mean variability

Radar precipitation distributions at 2830 m asl on the regional domain (450 m resolution, Fig. 5) show a larger interquartile range (IQR) than radar precipitation on the local domain (300 m resolution, Supplementary Information S2), confirming that local precipitation is more uniform than regional precipitation. Radar median precipitation over 24 h is on the order of 10 to 20 mm water equivalent for all three precipitation events in the regional domain. The median of radar precipitation in the

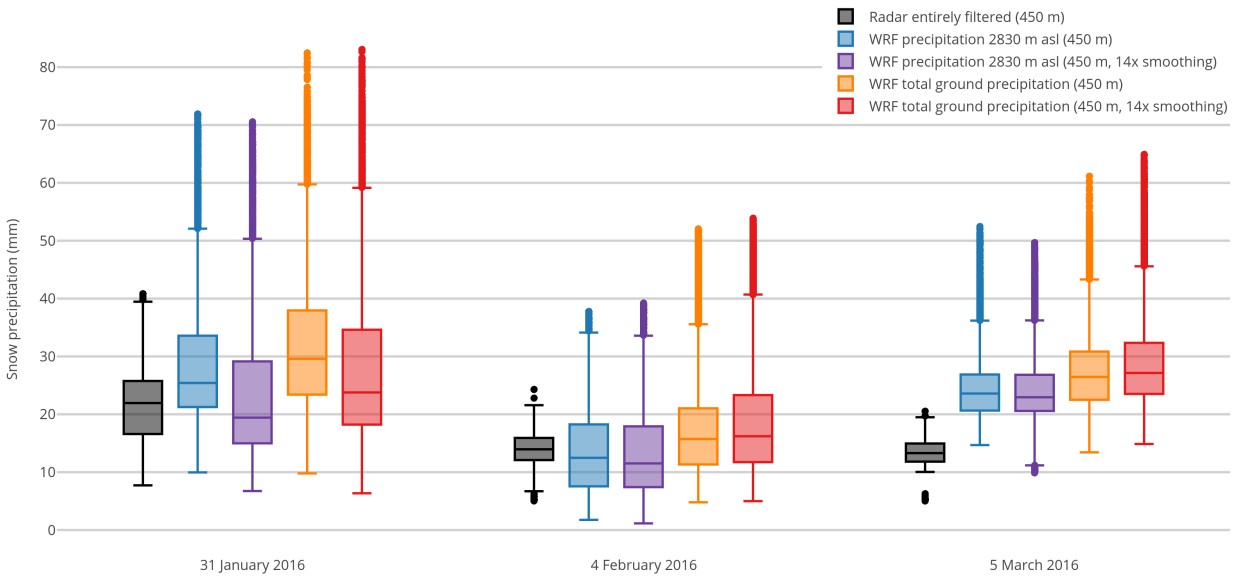

**Figure 5.** Domain-wide 24 h precipitation statistics for the regional domain (450 m resolution, Fig. 1) for the three precipitation events on 31 January 2016, 4 February 2016 and 5 March 2016. Gray colors show entirely-filtered radar precipitation. WRF precipitation at 2830 m above sea level (m asl) for simulations with weak terrain smoothing (Sect. 2.1) and strong terrain smoothing are given in blue and violet, respectively. Orange (red) shows boxplots of WRF total ground precipitation for weak (strong) terrain smoothing. Radar precipitation and WRF precipitation at 2830 m asl are masked (as shown in Fig. 4).

local domain can be both, higher or lower than in the regional domain. Although radar estimates are based on a reference S-Z relationship, the employed formula (Eq. 1) is not immune to potential estimation errors. Therefore, despite reasonably assuming that the potential estimation errors should not significantly influence the variability and the relative intensity of the precipitation fields, we consider potential inaccuracies in our interpretations.

5     For the precipitation events on 31 January 2016 and 4 February 2016 median precipitation at 2830 m asl in the COSMO–WRF simulations is in reasonable agreement with radar median precipitation (Fig. 5), even though WRF and radar precipitation patterns are different (Sect. 3.2). However, for the precipitation event on 5 March 2016 the median of precipitation in the regional domain is higher in WRF simulations compared to radar measurements, while the large-scale precipitation gradient is in good agreement (Fig. 4c and Fig. 4f). The IQR of WRF precipitation is generally larger compared to the IQR of radar precipi-
10 tation and the domain-wide WRF precipitation distribution has longer tails compared to the radar precipitation distribution. On the local domain these tendencies are preserved (Supplementary Information S2). Median precipitation is higher in the WRF

simulations compared to radar estimates on 31 January 2016 in addition to 5 March 2016, for which the deviations increase. This supports the hypothesis that the model tends to overestimate precipitation for higher resolutions with steeper and more complex topography.

The domain-wide median and IQR of precipitation at 2830 m asl in WRF simulations with weaker and stronger terrain smoothing (Sect. 2.1) are similar with a slight tendency of higher median values for weaker smoothing, indicating that the accuracy of topography does not have a strong influence on the domain-wide statistics of precipitation on the regional scale. Enhanced precipitation for weaker terrain smoothing compared to stronger terrain smoothing could be explained by enhanced precipitation production due to steeper topography.

An overestimation of precipitation in WRF simulations was previously reported (e.g. Mass et al., 2002; Leung and Qian, 2003; Silverman et al., 2013) and could be due to various reasons. Mass et al. (2002) and Leung and Qian (2003) among others report that WRF tends to show a stronger overestimation of precipitation for higher model resolutions compared to coarser model resolutions. Additionally, they document a dependency on the intensity of precipitation. An overestimation of orographic precipitation enhancement in more complex terrain or an overestimation of moisture in the model were further reported by Silverman et al. (2013). An overestimation of orographic precipitation enhancement would be in agreement with a stronger overestimation of precipitation for the local domain compared to the regional domain and for weaker smoothing compared to stronger smoothing. Furthermore, it is likely to occur for simulations with high horizontal resolution as higher peaks and steeper slopes are preserved (Silverman et al., 2013). Compared to a shallow topography, higher peaks and steeper slopes may cause stronger lifting and subsidence, which is also a likely cause for additional drops in relative humidity in WRF compared to measured relative humidity (Sect. 3.1, Fig. 2 and Fig. 3). This tendency seems to only apply for the highest elevations. For lower elevations strong smoothing may result in elevation differences, which are too small for precipitation to evolve by lifting condensation (Sect. 3.2). As additional reasons for precipitation overestimation in WRF an overestimation of precipitation in the driving model (Caldwell et al., 2009) and underlying landuse characteristics (Silverman et al., 2013) were mentioned. The latter was, however, previously found to only have a weak influence on the precipitation amount (Pohl, 2011). Humidity in COSMO–2 is an unlikely reason as COSMO–2 shows rather a tendency of underestimating relative humidity compared to station measurements (Fig. 2 and Fig. 3). Even though there are many possible reasons for an overestimation of precipitation in WRF, the estimation of solid precipitation from radar measurements is also subject to uncertainties (e.g. Cooper et al., 2017). Given uncertainties in radar precipitation estimates the comparison of median domain-wide precipitation should be taken with care. An in depth analysis of spatial variabilities is given in Sect. 3.4.

At the ground level WRF precipitation tends to show higher median values of precipitation compared to WRF precipitation at 2830 m asl for both domains. The IQR is similar. From this we hypothesize that there are precipitation formation or enhancement processes taking place between the elevation of the radar and the ground. This is in good agreement with the fact that near-surface processes can strongly enhance snow precipitation (e.g. riming). Overall, this analysis shows that WRF tends to overestimate domain-wide precipitation and precipitation variability at 2830 m asl compared to radar estimates.

### 3.4 Spatial variability

To address spatial patterns and variability of precipitation a scale analysis is performed augmented with a 2D-autocorrelation analysis (Sect. 2.4). Given the overestimation of precipitation in the model and the large differences in domain-wide variability between the model and radar precipitation estimates (Sect. 3.3), all variograms are normalized by the domain-wide variability, which allows analysis of spatial patterns with respect to the overall range of precipitation values. From the analysis of precipitation patterns (Sect. 3.2), we further know that there are strong large-scale precipitation gradients in the regional domain. In the variogram analysis small- and intermediate-scale structures may be hidden by the large-scale gradient. To avoid this and non-stationarity of patterns, we first present variograms of detrended precipitation fields (Sect. 3.4.2). However, to assess processes acting at different scales, variograms of non-detrended precipitation patterns are subsequently analyzed in a scale analysis (Sect. 3.4.3). Finally, a 2D-autocorrelation analysis is used to comment on directional dependencies of precipitation patterns (Sect. 3.4.4).

### 3.4.1 Large-scale precipitation trends

Large-scale precipitation patterns show a strong gradient (Fig. 4). Therefore a plane is fit linearly to the precipitation fields describing the large-scale precipitation trend (Table 2). The trend on the 31 January 2016 roughly points toward the North. For the precipitation event on 5 March 2016 the trend points roughly to the South. Given a southerly advection on 5 March 2016 this direction corresponds to the main wind direction and therefore agrees with the theory of large-scale orographic precipitation enhancement or rather the drying trend due to sinking air further downstream within the mountain range. The south-north gradient on 31 January 2016 roughly agrees with the main wind direction but points out that regional trends of larger-scale patterns may not exactly be aligned with wind direction. For the precipitation event on 4 February 2016 the intensity of the trend (i.e. the strength of inclination of the linearly fitted plane) is, however, weak and therefore the orientation of the slope is arbitrary. For this day, we hypothesize that either dynamics were more variable preventing the evolution of a strong gradient or lifting condensation due to the orography was not as efficient as for the other two events. For two events (31 January 2016 and 4 February 2016) the model has trouble reproducing the trend (i.e. orientation and intensity of the linearly fitted plane). For 31 January 2016 the deviation of orientation between the trends of radar precipitation and WRF precipitation at 2830 m asl is about $70°$ but with a similar intensity of the trend. For 4 February 2016 the model shows a strong trend of precipitation, while the intensity of the trend is weak in the entirely-filtered radar data. However, for the precipitation event on 5 March 2016 the trend is reasonably captured by the model with a deviation of the orientation of $16.6°$ and a slightly stronger intensity of the trend in the model compared to the radar estimation.

Disagreement in precipitation patterns, trend orientation and intensity on 4 February 2016 (quite homogeneous precipitation distribution in the radar estimate (Fig. 4) compared to the strong downstream shift of precipitation in WRF) and the overestimation of precipitation in the model give evidence for a too simplistic representation of precipitation in the model (i.e. simplified microphysics and cloud dynamics), which tends to overestimate the effect of highest topographic features but misses precipitation over shallower areas. Good agreement in the intensity of the trend on 31 January 2016 and good agreement of

**Table 2.** Large-scale linear trends of entirely-filtered radar and WRF precipitation patterns on the regional domain (450 m horizontal grid spacing, Fig. 1). WRF precip. at 2830 m asl refers to solid precipitation in WRF simulations at 2830 m above sea level and WRF total ground precip. refers to the total (solid and liquid) precipitation at the ground level. *Orientation* gives the direction of the slope and *Intensity* the strength of inclination. 0° would indicate a slope pointing toward the East. WRF snow precipitation is from simulations with weak terrain smoothing (Sect. 2.1).

|  | 31 January 2016 | | 4 February 2016 | | 5 March 2016 | |
| --- | --- | --- | --- | --- | --- | --- |
|  | Orientation | Intensity | Orientation | Intensity | Orientation | Intensity |
| Radar entirely filtered | 86.9° | 0.17 | -125.9° | 0.01 | -114.8° | 0.04 |
| WRF precip. at 2830 m asl | 16.7° | 0.22 | -5.1° | 0.21 | -98.2° | 0.12 |
| WRF total ground precip. | 25.0° | 0.18 | 5.4° | 0.26 | -103.3 | 0.19 |

the orientation of the trend on 5 March 2016, however, show that the model is able to capture large-scale precipitation trends, which may be connected to a large-scale orographic enhancement.

### 3.4.2 Spatial variability of detrended precipitation fields

On the smallest scales a strong difference is visible in variograms of detrended entirely-filtered and detrended partly-filtered radar precipitation, with weaker variability for entirely-filtered data (Fig. 6). Smallest-scale structures in the radar data are likely an indicator of residual noise in the partly-filtered radar data (Sect. 2.3). However, it could also imply micro-scale precipitation features. This stresses the challenge of processing high-resolution radar data (Sect. 2.3) to get a reasonable radar precipitation field. In any case the entirely-filtered radar precipitation estimates may be regarded as clean concerning residual clutter and will therefore be used for all subsequent analysis.

Variograms of entirely-filtered and detrended radar precipitation show a steep increase of variability on the smallest scales, while the increase in variability gets weaker for larger scales (less steep slope in the variograms). Small-scale patterns are likely driven by small-scale precipitation cells induced by local cloud dynamics and microphysics. Such small-scale structures are repeated on intermediate scales and lead to a weaker increase in variability, as less new spatial features are added. At larger scales variability reaches the total variability of the detrended data.

Compared to radar precipitation WRF precipitation at 2830 m asl shows a lower variability and a flatter increase in variability at small scales giving evidence for a smoother precipitation distribution at the smallest scales. The lack of small-scale patterns shows that the radar sees more variability at the smallest scales, while WRF likely misses the smallest-scale processes. Variability of radar and WRF precipitation at 2830 m asl at large scales (> 5 km), especially on 4 February 2016, show less systematic differences than at small scales. This indicates that, with respect to total variability, patterns at these scales are well represented. Total ground precipitation shows a higher variability compared to precipitation at 2830 m asl (except for 4 February 2016), which is an indication that near-surface processes are active in the model.

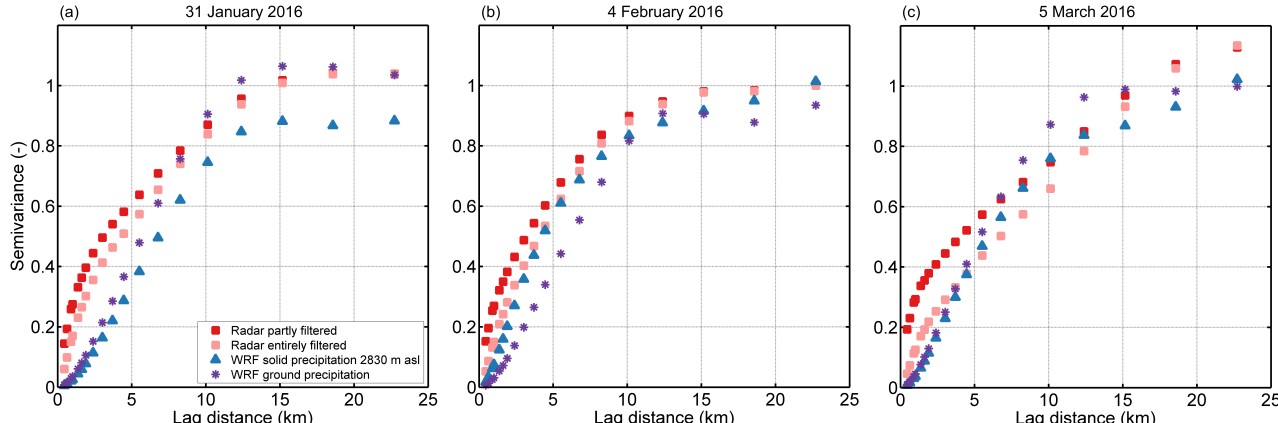

**Figure 6.** Variograms of detrended snow precipitation normalized by the domain-wide variance of precipitation for the precipitation events on a) 31 January 2016, b) 4 February 2016 and c) 5 March 2016 for the regional domain (450 m horizontal grid spacing, Fig. 1). Variograms are given for partly-filtered (red) and entirely-filtered (orange) radar snow precipitation, WRF snow precipitation at 2830 m above sea level (m asl, blue) and WRF total ground precipitation (violet). WRF precipitation is from simulations with weak terrain smoothing (Sect. 2.1). All precipitation fields are masked.

Variograms of precipitation in the local domain (300 m resolution, Fig. 1) look similar to variograms of the regional domain (450 m reoslution), but reach domain-wide variability at about 5 km lag distance (Supplementary Information S2), while on the regional scale the domain-wide variability is reached at a distance of about 15–20 km (Fig. 6). Furthermore, the difference between radar and WRF precipitation variability at small scales is larger on the local domain compared to the regional domain.
This and a systematic underestimation of precipitation variability at scales < 5 km (on the regional domain) compared to the precipitation variability in radar estimates indicate that mountain ridge-scale precipitation processes are under-represented in the model.

### 3.4.3   Scale breaks and dominating processes

Scale breaks were previously found to be connected to changes in dominant processes (e.g. Deems et al., 2006). Here, we
present a scale analysis including variability due to large-scale precipitation processes. Therefore, we present variograms of non-detrended precipitation fields, being aware that a certain portion of the small- and intermediate-scale precipitation variability may become hidden. As precipitation patterns are known to be driven by topography and wind, we present variograms of topography together with the variograms of precipitation.

Variograms of topography clearly reveal two scale breaks (Fig. 7). The first scale break is between 1 and 2.5 km depending
on the resolution, the second scale break is at 5 km and 6 km for real topography and weakly smoothed WRF topography, respectively. For topography the two scale breaks are separating the mountain-slope scale (< 1–2 km), mountain-ridge-to-valley scale (between ∼ 1–2 km and ∼ 5 km) and the scale of repeated mountain ridges and valleys (> 5 km).

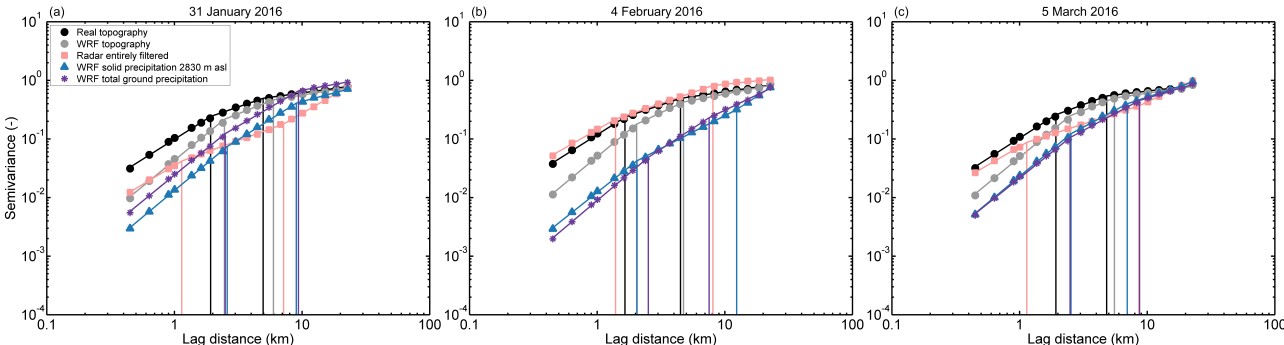

**Figure 7.** Normalized variograms of the snow precipitation events on a) 31 January 2016, b) 4 February 2016 and c) 5 March 2016 for the regional domain (450 m horizontal grid spacing, Fig. 1). Variograms are given for entirely-filtered radar snow precipitation (orange), WRF snow precipitation at 2830 m above sea level (m asl, blue) and WRF total ground precipitation (violet). Additionally, variograms are given for real topography (based on dhm25 © 2018 swisstopo (5740 000 000), black) and WRF topography (gray). WRF topography and precipitation are from simulations with weak terrain smoothing (Sect. 2.1). All precipitation fields are masked.

For consistency reasons, all variograms in Fig. 7 are presented with two scale breaks. Scale breaks for all events and both resolutions are basically grouped in two areas (∼ 1–2.5 km and 5–10 km for 450 m resolution, Fig. 7 and ∼ 800 m–1.2 km and 2.5–5 km for 300 m resolution, Supplementary Information S2), even though for precipitation some scale breaks are arbitrary. Albeit scale breaks of precipitation do not exactly match scale breaks of topography, breaks at similar scales as well as similar slopes of topography and precipitation at small scales support the interpretation of topography dependent precipitation patterns. On the smallest scales (< 1–2 km) the slopes of precipitation variograms are similar to the slopes of the variograms of corresponding topography. This is an indication that precipitation patterns on mountain-slope scales may be terrain-driven. Processes acting at these scales could be small-scale cloud-dynamical processes such as the seeder-feeder mechanism (Bergeron, 1965; Purdy et al., 2005) or preferential deposition (Lehning et al., 2008). The latter is, however, for most mountain ridges unlikely to be seen in precipitation fields at 2830 m asl as it happens close to the ground. For the precipitation event on 5 March 2016, on scales > 5–7 km (i.e. for the scales above the second scale break) the slopes of the normalized variograms of radar and WRF precipitation at radar elevation are similar. Large-scale gradients at these scales are most likely driven by large-scale orographic precipitation enhancement (e.g. Stoelinga et al., 2013). Good agreement of the slopes in normalized variograms between radar and WRF precipitation is an indicator that the model has the potential to properly represent the strength of the large-scale gradient with respect to the overall variability, i.e. large-scale orographic precipitation enhancement. Disagreement of variograms of precipitation and topography at these scales further support the hypothesis that largest-scale precipitation is mainly determined by large-scale orographic precipitation enhancement, which introduces an increase in variability of precipitation at large scales, while large-scale topography reveals a repeated pattern of valleys and peaks (i.e. constant variability). Overall, this analysis supports the hypothesis in Sect. 3.2 that precipitation patterns in the regional domain are topography driven.

### 3.4.4  2-dimensional variability patterns

Finally, the combined influence of topography and the general wind direction on snow precipitation patterns in the regional domain is assessed by spatial 2D-autocorrelation maps (Fig. 8). Like variograms, autocorrelation is dependent on large-scale trends. The general direction of 2D-autocorrelation patterns is the same for detrended (Fig. 8) and non-detrended (not shown) precipitation patterns. However, autocorrelation patterns of detrended precipitation fields show much shorter decorrelation lengths. This is due to the spatial coherence introduced by large-scale trends in precipitation. To avoid biased autocorrelation data, only 2D-autocorrelation maps of detrended precipitation fields are shown. However, we keep in mind that large-scale trends are present.

Autocorrelation maps of topography (Fig. 8a and Fig. 8e) represent a north-west to south-east oriented pattern which is, although weaker, repeated in south-west to north-east and west to east direction. For snow precipitation events with dominating north-westerly to northerly advection, the main axis of the snow precipitation 2D-autocorrelation pattern is oriented in a north-west to south-east direction and therefore in alignment with both topography and the main wind direction (Fig. 8b-c and Fig. 8f-g). Patterns of WRF precipitation at 2830 m asl are rotated toward a north-south direction on 4 February 2016. For dominating southerly advection the 2D-autocorrelation map of radar precipitation shows a more homogeneous pattern compared to autocorrelation patterns for northern to north-western advection but a weak south-west to north-east orientation of larger-scale patterns (Fig. 8d). For the WRF simulations a strong south-west to north-east orientation is present in the autocorrelation map for the precipitation event on 5 March 2016 (Fig. 8h). Even though isotropic variograms reveal good agreement in domain-wide variability, 2D-autocorrelation maps show that this may not necessarily go along with good agreement of the orientation of patterns. Best agreement in the orientation of patterns is found for 31 January 2016. For the three events, 2D-autocorrelation maps of detrended precipitation reveal a smoother distribution of precipitation on the smallest scales in the model compared to radar data, due to less small-scale structures in the model. On the other hand, a strong decrease in autocorrelation in east-west direction is visible for 5 March 2016. This shows that WRF simulations have a stronger dependency on both wind direction and topography and tend to generate strong precipitation bands in the main wind direction, confirming the overly simplistic behavior of the model.

For ground precipitation 2D-autocorrelation patterns tend to be repeated in south-west to north-east and west to east direction as seen for topography (Figs. 8j-g). This stresses the hypothesis that the influence of topographic features on WRF ground precipitation is stronger than at radar elevation and gives evidence that these results are likely produced by near-surface topographically driven pre-depositional processes such as e.g. preferential deposition or the seeder-feeder mechanism in the model. While a topography dependency was already found in isotropic variograms, this 2D-autocorrelation analysis reveals that the wind direction additionally strongly impacts the snow precipitation distribution.

### 3.5  Dependence of spatial variability on model resolution and smoothing

Geostatistical analyses presented in this study demonstrate that precipitation on the regional scale ($> 5$ km) is reasonably represented in the WRF model, while small-scale precipitation variability is systematically underestimated in the model sim-

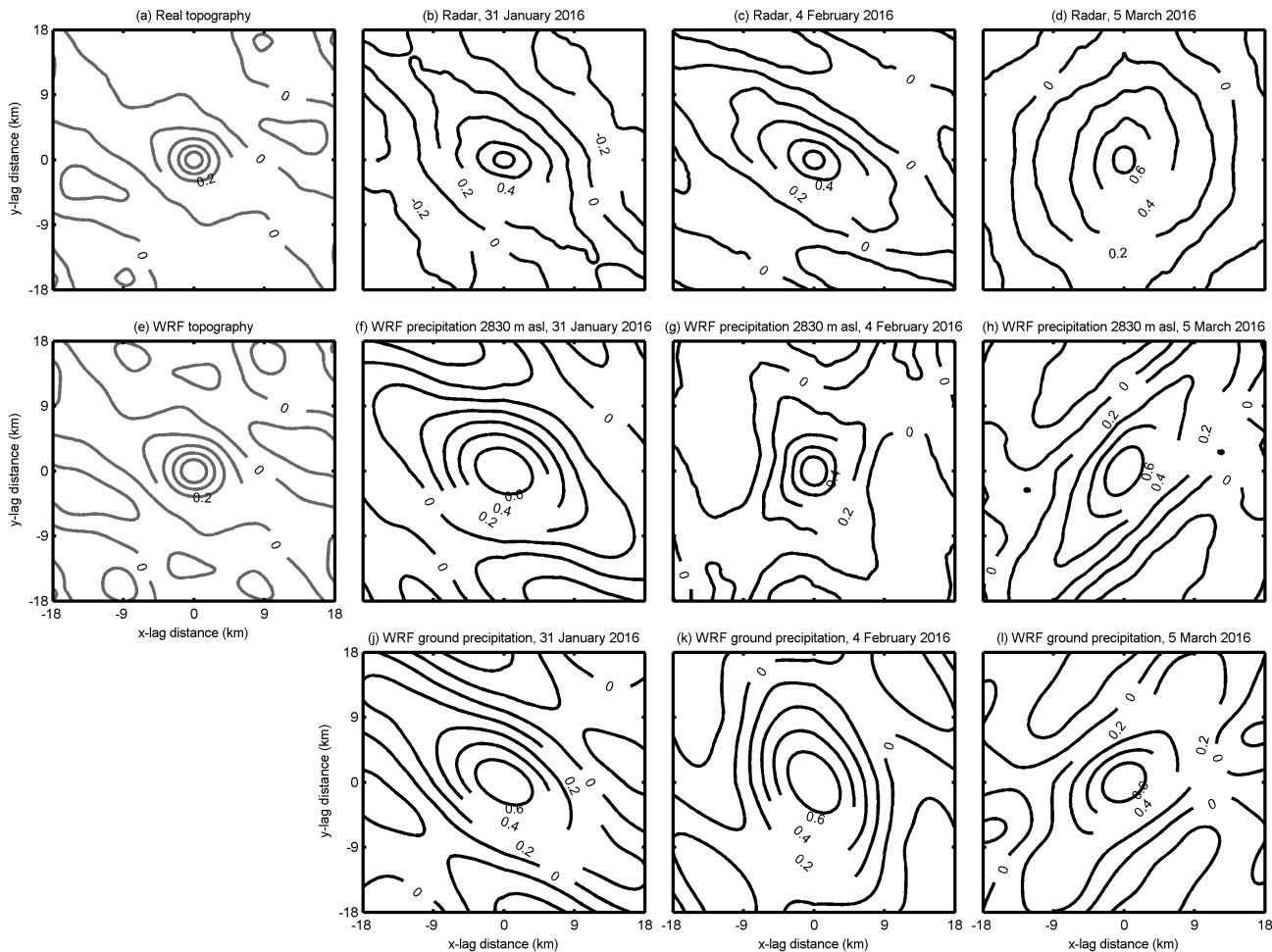

**Figure 8.** Spatial 2D-autocorrelation maps for the regional domain (450 m resolution) of detrended a) real topography (based on dhm25 © 2018 swisstopo (5740 000 000)), b)-d) entirely-filtered radar snow precipitation, e) WRF topography, f)-h) WRF snow precipitation at 2830 m above sea level (m asl) and j-l) WRF total ground precipitation. Autocorrelation maps of snow precipitation are for the three snow precipitation events on 31 January 2016, 4 February 2016 and 5 March 2016. WRF topography and precipitation are from simulations with weak terrain smoothing (Sect. 2.1). Radar precipitation and WRF precipitation at 2830 m asl are masked (as shown in Fig. 4).

ulations with a horizontal grid spacing of 450 m (Sect. 3.4). Variograms up to a maximum lag distance of 5 km on domain Dischma (Fig. 1) reveal an increase of variability for increasing model resolution (Fig. 9). However, simulated variability stays far below the variability of entirely-filtered radar precipitation. Depending on the event an increase in variability is present for 150 m and 50 m resolution. This indicates that smallest-scale precipitation dynamics are still not fully resolved at 50 m resolu-

5    tion. A comparison of variograms for simulations with strongly smoothed terrain compared to simulations with weaker terrain

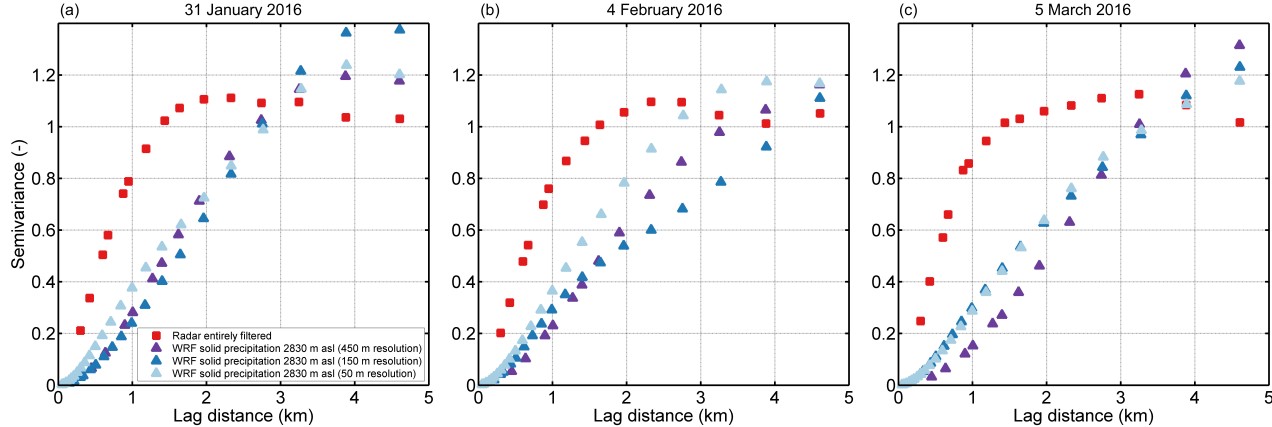

**Figure 9.** Variograms of detrended snow precipitation normalized by the domain-wide variance of precipitation for the precipitation events on a) 31 January 2016, b) 4 February 2016 and c) 5 March 2016 for domain Dischma (Fig. 1). Variograms are given for entirely-filtered radar snow precipitation (red), and WRF snow precipitation at 2830 m above sea level (m asl) with 450 m (violet), 150 m (blue) and 50 m (light blue) resolution. WRF snow precipitation is from simulations with weak terrain smoothing (Sect. 2.1). Radar precipitation is masked.

smoothing (Sect. 2.1) reveal that a stronger terrain smoothing may result in less explained variability in normalized variograms (not shown). Even though this signal is not consistent for all events, we can show that a better representation of topography due to higher resolution and less smoothing has the potential to increase the explained variability of precipitation patterns. An increase in variability at small scales (< 5 km), indicates that more small-scale patterns are resolved at higher resolutions (50 m horizontal grid spacing) in the model. Our simulations are currently limited to the presented resolutions and strong terrain smoothing due to model instabilities. However, based on the presented results, once available, the immersed boundary method version of WRF (e.g. Lundquist et al., 2010, 2012; Arthur et al., 2016; Ma and Liu, 2017), will likely be a good tool to allow for steeper slopes in the simulation and going toward higher resolution LES simulations to resolve further small-scale wind fields, which drive the precipitation structures.

## 4  Conclusions and Outlook

The implementation of COSMO–WRF is a further step in performing very-high resolution precipitation simulations in complex alpine terrain to address the question of the relative importance of cloud-dynamics and particle-flow interactions on a mountain-ridge scale. In this validation study, we show that COSMO–WRF is able to reasonably simulate temperature, relative humidity and wind direction, but tends to (strongly) overestimate near-surface wind speeds. This may be due to many reasons from an overestimation of speed-up effects on the windward side of mountain ridges to an underrepresentation of small terrain features. Additional reasons are likely but remain unknown and future work is needed to address these issues. Relative humidity patterns

are highly variable and may be a sign that subsidence and lifting produce too strong effects in the (partially parameterized) cloud dynamics, given the good representation of topography at larger scales.

Regional and local scale precipitation patterns in the COSMO–WRF simulations are in partially good agreement with MeteoSwiss operational radar measurements and automatic weather stations. For the three events analyzed here, precipitation estimates from WRF simulations are higher compared to precipitation estimates from radar measurements. A general overestimation of precipitation produced by WRF is consistent with an overestimation of subsidence and lifting. Overestimation of precipitation in WRF simulations has been documented previously for snow precipitation over complex terrain (e.g. Silverman et al., 2013), among other reasons it may be due to the high model resolution and therefore more complex topography and higher mountain peaks compared to common high resolution simulations.

An autocorrelation and scale analysis of radar and WRF snow precipitation reveals a good agreement of precipitation patterns on regional scales (> 5 km), which are topography and wind driven. These large-scale patterns are in good agreement with the theory of large-scale orographic enhancement (e.g. Stoelinga et al., 2013). Disagreement in precipitation patterns i.e. a downwind shift of snow precipitation in the WRF simulations compared to radar precipitation estimates is likely due to lifting condensation being too weak in areas, where topography is lower and strong smoothing leads to an underrepresentation of topography. On the other hand, over peaks, which are high and steep enough in the model to allow for lifting condensation, the effect of orographic precipitation enhancement tends to be overestimated. An increase of this overestimation of precipitation over high elevations for higher resolution simulations as well as for weaker terrain smoothing supports this hypothesis. Smallest-scale patterns in the radar measurements are likely dominated by noise, which is removed by the application of a median filter. Given these uncertainties the radar data cannot be considered as the absolute reference. In case of critical data analysis, an estimation of high-resolution radar precipitation is, however, useful to improve the understanding of precipitation processes in complex terrain and to validate and improve model simulations. On a local to mountain-valley scale WRF simulations systematically show a lower variability of precipitation compared to radar estimates. This indicates that the model is not able to represent the full spectrum of small-scale precipitation patterns, which are present in the radar measurements. One potential reason for the lack of precipitation variability is the simplification of cloud dynamics and microphysics in the model, typically used to model regional-scale precipitation fields. Additionally, we could show that the underrepresentation of topography may have a strong influence on the formation of local low-level clouds, which are important for orographic precipitation enhancement. This is supported by the fact that precipitation patterns in the model show a stronger dependency on topography and wind direction than precipitation patterns in the radar estimates. However, an increase in precipitation variability at scales < 5 km is visible for higher resolution WRF simulations. Furthermore, for simulations with steeper terrain an increase in variability for all resolutions is found. This shows that especially for small-scale variability a better representation of the complex terrain is essential to reproduce precipitation variability. Although the model cannot represent the full variability measured by the radar at small scales, an increase in precipitation between 2830 m asl and the ground is an indication that the model captures a certain portion of near-surface processes.

To specifically address processes such as the seeder-feeder mechanism or preferential deposition an analysis of hydrometeors and precipitation distributions in vertical profiles across mountain ridges is needed. To connect pre-depositional processes with

post-depositional processes even higher resolution WRF simulations would be required. This might be achieved by employing the immersed-boundary method version setup of WRF. A parameterization of post-depositional processes in WRF or using WRF simulations as a boundary condition for simulations with the Alpine surface processes model Alpine3D (Lehning et al., 2008), would then allow validation of modeled snow accumulation patterns compared to measured snow accumulation patterns. Furthermore, simulations of precipitation patterns in complex terrain need to be analyzed with higher temporal resolution (e.g. on the order of minutes), as contributing processes show high temporal variability. Future work will include addressing the temporal variability of precipitation patterns using radar observations, along with an analysis of precipitation growth with respect to topography and wind direction.

*Code and data availability.* A COSMO–WRF documentation is published as Gerber and Sharma (2018). Data can be made available upon request.

## Appendix A: Morrison microphysics in WRF

The Morrison microphysics scheme includes prognostic equations of number concentration and mass mixing ratio of 5 precipitation species (rain, snow, ice, graupel and cloud droplets). The parametrization of rain, snow, ice and cloud droplets is based on Morrison et al. (2005). The implementation of graupel follows Reisner et al. (1998), except for minimum mixing ratios, which are required to produce graupel from the collision of rain and snow, snow and cloud water, and rain and cloud ice, which are based on Rutledge and Hobbs (1984).

The kinetic equations include advection, sedimentation and turbulent diffusion as well as source and sink terms of ice nucleation and droplet activation, condensation and deposition, coalescence and diffusional growth, collection, melting and freezeing as well as ice multiplication (Morrison et al., 2005). For graupel deposition, collection, collision, accretion, freezing and melting processes are parameterized (Reisner et al., 1998).

Size distribution functions are gamma functions:

$$N(D) = N_0 D^\mu e^{-\lambda D}, \tag{A1}$$

where D is the particle diameter, $\mu$ is the shape parameter of the distribution function, which is $\mu = 0$ for rain, snow, ice and graupel, resulting in an exponential function for N(D). $\lambda$ and $N_0$ are the slope and intercept, respectively, of the size distribution, evaluated by the predicted number concentration $N$ and mass mixing ratio $q$:

$$\lambda = [\frac{cN\Gamma(\mu+d+1)}{q\Gamma(\mu+1)}]^{1/d} \tag{A2}$$

and

$$N_0 = \frac{N\lambda^{\mu+1}}{\Gamma(\mu+1)}, \tag{A3}$$

where $\Gamma$ is the gamma-function. $c$ and $d$ are the parameters of the power-law function $m = cD^d$ indicating the mass-diameter relationship. Terminal fallspeeds are as well assumed to have a power-law form of $v(D) = \frac{\rho_{sur}}{\rho} aD^b$, with individual parameters $a$ and $b$ for the different species. $\rho$ is the air density and $\rho_{sur}$ the air density at sea level. For simplification all species are assumed to be spheres. Additionally, the particles do not have any particle inertia.

*Author contributions.*  FG and NB performed the analysis, which was supported by fruitful discussions with ML, RM, AB and UG. FG, VS, MD, RM and ML developed the COMSO-WRF coupling and simulation setup, which were run by FG. NB and MG processed the radar data. FG, ML, RM, NB and AB contributed to the design of the concept. FG and NB with contributions of all authors prepared the manuscript.

*Acknowledgements.*  The work is funded by the Swiss National Science Foundation (Project: Snow-atmosphere interactions driving snow ac-cumulation and ablation in an alpine catchment: The Dischma Experiment; SNF-grant: 200021_150146 and project: The sensitivity of very
small glaciers to micrometeorology; SNF-grant: P300P2_164644). Topographic data are reproduced by permission of swisstopo (JA100118). For advice to setup WRF simulations we thank the WRF-help. For supporting our project with computational time many thanks go the the Swiss National Supercomputing Center (CSCS) and their support for technical advice. Additionally, we thank the Swiss Federal Office of Meteorology and Climatology (MeteoSwiss) for providing access to radar data, COSMO–2 analysis and the regridding tool fieldextra. For advice related to COSMO–2 thanks go to Guy de Morsier from MeteoSwiss. Further thanks go to Amalia Iriza and Rodica Dumitrache from
MeteoRomania for advice concerning the COSMO–WRF coupling. Additional thanks go to Louis Quéno and Benoit Gherardi for their work on pre-preprocessing steps and data processing to setup WRF simulations. Additionally, we thank Heini Wernli and the two anonymous reviewers for their questions, comments and recommendations, which helped to improve the paper.

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
