# Peer review of "Spatial variability of snow precipitation and accumulation in COSMO-WRF simulations and radar estimations over complex terrain"

_The Cryosphere, 2018_

## Referee Comment (RC1) · H. Wernli (Referee) · 14 Apr 2018

Review of paper

"Spatial variability of snow precipitation and accumulation in COSMO–WRF simulations and radar estimations over complex terrain"

by F. Gerber et al.

submitted to The Cryosphere

This paper presents a comparison of snow precipitation fields from very high-resolution WRF simulations and radar estimates for three recent snowfall events near Davos

(Switzerland). WRF simulations are performed in LES mode down to 50 m horizontal grid spacing. The technical challenges of this project are enormous and I congratulate the authors for doing this detailed analysis. Unfortunately, the WRF simulations of snow precipitation deviate rather strongly from radar observations (- I have a slightly more negative impression of the model performance than the authors because of the partially huge differences in near-surface winds and strong precipitation overestimations) and this model-observation differences make it challenging to learn a lot from these simulations. The authors use a set of statistical tools and in my impression the variogram analysis is particularly interesting and delivers the most valuable results of this study. I recommend to accept this study for publication with major revisions in order to improve the readability of this paper and to clarify a range of issues (see detailed comments below).

Major comments ————

A) I found it rather difficult to read the paper and follow the story-line. I very strongly recommend that the authors add a paragraph at the end of the introduction, where they clearly outline the specific objectives of the study and why they need the tools they use to address these objectives. In the current manuscript I was missing a clear roadmap: what are the objectives and questions? Why do we look at a certain diagnostic? What in the end did we learn from a specific section?

B) To further increase the readability, the text should be improved and several unclear statements amended. Several examples are given below in the minor comments. In several paragraphs I was left with a rather unclear impression of what I should have learned from reading the paragraph - a more reader-friendly writing style would help.

Minor comments ————

1) p. 1 line 3: the term "particle-flow interaction" appears several times in the paper but is never explained. I am not sure whether I understand the term, is it about how the turbulent flow influences the pathway of individual snow particles? Is this really the

topic of this study? Please clarify.

2) p. 1 line 5: it looks as if COSMO-WRF was the name of the reanalysis data, but this is not what you mean - please clarify.

3) p. 1 line 10: not clear what "variability relative to the domain wide variability" means

4) p. 1 line 11: "quite substantial" is not scientific language, can you be more specific?

5) p. 1 line 11: The last sentence of the abstract "However, ..." seems rather unconnected to the previous sentences

6) p. 2 line 7: here and in many other places: why are references not in chronological order?

7) p. 2 line 31 and p. 5 line 20ff: you refer to the COSMO-2 analysis fields as "reanalyses", which is most likely wrong. If I am not mistaken, then the fields you use are the operationally produced hourly analysis fields. The term REanalysis would only be appropriate if the older analyses were recalculated with a newer data assimilation approach. If this is the case, then it should be explained.

8) p. 3 line 1: I know what you want to say, but this is a strange sentence. Of course, relevant measurements can also be made with surface barometers, it just depends on what you want to measure!

9) p. 3 line 4: "orographic mechanisms": not clear to me what you mean

10) p. 3 line 20: here and in other places, use km for kilometer and m for meter

11) p. 3 line 33: mb should read hPa

12) p. 4 line 2: I would rather say that this is the northeastern part of Graubünden(see also p. 6 line 19); no need to write "(CH)"

13) p. 4 line 7: I find "mesoscale mode" a bit confusing, why not just LES and non-LES mode? And - see Table 1 - is there really no PBL scheme used in LES mode? I thought

that for the small-scale turbulence you still need a parameterization?

14) p. 5 line 3: what is a "combined simulation"?

15) p. 6 line 10: units (here K m-1) should not be in italics (see also several other places)

16) p. 7 line 31: should read "restrict ... to only four ..."

17) p. 8 line 21: I don't understand: "three different domains" but then four resolutions?

18) p. 8 line 27: please explain what a "lag distance bin" is and for what purpose you will use the semivariance

19) p. 10 last line: "For all stations ...": there are only two –> "For both stations ..."; then "matches well", hm, Fig. 2 shows deviations up to 6K, which is a lot!

20) Figure 2: I find the overestimation of winds in WRF for the first two cases rather dramatic (my impression is that this should be described more clearly); is there any reason why winds are so much better for the 3rd case?

21) p. 11 line 5: here I find the "COSMO blaming" a bit too strong, because for instance Fig. 2 shows that COSMO is doing better for the 2nd case. Rather often, the 2-m RH values are 100%, which indicates fog. Could it be that correctly simulating fog is a particular challenge for WRF?

22) p. 12 line 5: "For some stations ...", but you only discuss two!

23) p. 12 line 10: "could partially reduce overestimated wind ...": unclear, reduce compared to what?

24) p. 12 lines 12 and 20: "reasons ... may be manifold" is not a strong statement (and it should not be repeated). It seems that the "reasons are unknown", which is fine but it should be stated maybe in this way.

25) p 12 line 18: but if very small-scale topography matters so much, why then is

COSMO performing better for some cases?

26) p. 12 line 25: improve writing "in precipitation is observed for precipitation ..."

27) p. 12 line 27: The sentence "Although ..." is too long and very difficult to understand, in particular, why does the sentence start with "Although"?

28) p. 12 lines 28 and 34: I am a bit lost, aren't these two sentences contradicting? Once southerly flow leads to a N-S gradient and then for southerly advection there is a S-N gradient ...

29) p. 13 line 4: Another example where reading/understanding is difficult: "Disagreement of patterns ...": you probably mean "between WRF and radar", and then "strong smoothing of topography": you probably mean "in the model". Please improve the writing to make this type of sentences much clearer.

30) p. 13 line 7: "precipitation fed to domain d02" but before, on p. 5 line 4, you wrote that "hydrometeors cannot be used as a boundary condition in WRF simulations": how does this go together? What do you mean by "feeding precipitation"?

31) p. 13 line 11: you show the overestimation of winds only at 4-5m! It is not evident that winds are also too strong higher above.

32) p. 13 line 19: I don't understand partially filtered vs. median-filtered

33) Figure 4: I find it unfortunate that the different panels for the same case have different scales. Also, I find it difficult to interpret the topography contours: for non-locals it is not clear where valley and mountains are. A color plot of topography only could help.

34) Figure 5: I don't understand the difference between the 300 and 450m resolutions: why is this so important? why has it such a large effect? and why is this effect different for radar than WRF for the first case?

35) p. 15 line 2: Hm, isn't this a complicated way of saying that the median is higher if

there is more precipitation?

36) p. 15 line 3: what is meant by "shifted"? In WRF relative to radar?

37) p. 16 line 15: I don't understand "strong smoothing may not allow for precipitation to evolve"

38) p. 17 line 21 (and in other places): not sure that "idealistic" is the right term, do you mean "simplistic"?

39) p. 17 line 22: what is "particle dynamics"?

40) Figure 6: I first overlooked the small 10ˆ4 beneath the x-axis and wondered about the values. Why not just write ticks at 5, 10, ... km?

41) p. 22: to me, these two statements are contradicting each other: first you say that topographic features are underrepresented (I agree) and then that subsidence and lifting are too strong - but aren't they linked to topography?

42) p. 23 line 17: not sure that I understand "When observing with a critical eye", what is observed here?

43) p. 24 line 2: I don't understand this important sentence: I somehow understand that you say that WRF has too simplified cloud microphysics, but then why "due to the given resolution"? Do you say that the microphysics is OK for another resolution?

---

## Referee Comment (RC2) · Anonymous Referee #2 · 1 May 2018

General considerations

In this paper, the authors use a COSMO-WRF dynamical downscaling suite (one-way nesting) to investigate the ability of the numerical model to reproduce the spatial variability of precipitation in complex mountainous terrain. The authors have a very valuable (and quite novel) data set at hand for their analysis and perform model simulations, which indeed are able to address the raised questions. The paper is well presented (but see the quite numerous 'detailed comments'), the analysis is sound and the authors do, in general, appropriately value their results and critically assess them. Even if I have two comments that are termed 'major', they will easily be addressed, so that

my overall disposition is 'minor revisions'.

Major comments

1) Explanation of WRF offsets (Section 3.1, point inter-comparison). It is argued that these offsets were (partly) 'due to offsets in the COSMO-2 input' (p11, l. 2/5; p12, l. 11). This is a weak argument. Either the domain and time settings (e.g., distance from the inflow domain border; spin-up time) are chosen in a way that allows the downscaled model run to develop its own local conditions - which would demonstrate the improvement (or deterioration) due to the higher resolution, or the downscaling is essentially useless. This argument should be re-considered. Furthermore, for wind speed in particular, e.g. Jimenez and Dudhia (2012) come to a quite different conclusion than the present authors (but based on simulations for which the roughness length is not artificially enhanced). The present results should be at least discussed in the light of their results. Also, it should be noted that apparently WRF can produce some 8 m/s wind speed at times, while the observation is around 2 m/s or even less. On other days (different wind direction!) the WRF bias is not nearly as large. This does not seem to be a problem of mere roughness length or some problem in the 'model physics'. To me, this rather looks like a problem with the height attribution (or 'correction' - using a logarithmic [neutral?] profile - of the modelled profile to the observation height). Could it be that the actual observation height was '2 m minus snow depth' (rather than 2 m as in the model)? So, in case of a deep snow layer at the observational site the actual observation height would possibly be much smaller. In any case, this issue should be analysed in some more detail.

2) Overall conclusions: the authors nicely demonstrate the impact of topography (in different model resolutions) on the quality of the spatial variability of precipitation estimates by the model. In the conclusions, however, the arguments are somehow mixed up. First, (p24, l.2) it is argued that the difficulty in reproducing the small-scale variability is 'most likely [due to] the representation of cloud dynamics and microphysics in the model [being] too idealistic'. This is then supported by a number of arguments, all refer-

ring to the topography (...). Then, some few lines later (p24, l. 6) it is concluded 'This shows that especially for small-scale variability a better representation of the complex terrain is essential to reproduce precipitation variability'. So, what is it now? The terrain representation or the microphysics being to idealized? In my view, the results of the authors have demonstrated the impact of terrain (and resolution), but no conclusions can be drawn concerning the microphysics and cloud dynamics. Certainly, the authors have not done that – and so, such a 'conclusion' concerning microphysics and cloud dynamics cannot be maintained.

Minor comments

P3, l.1 means, not mean

P3, l. 33 hPa, not mb

P4, l. 4 set up, not setup

P4, l. 4 with refined vertical levels: from the above I understand that also d3 has refined vertical levels (60). Does this sentence mean, that these are not within the boundary layer? Please explain.

P4, l.6 nests run in 'LES'-mode: for d2 (450 m horizontal grid spacing) and possibly d3 (150 m) this is in the middle of the 'Gray Zone' of Wyngaard (2004). Can the authors comment on that?

Tab 1 please add the heights of the first few model levels

P6, l.11 logarithmic wind profile: does this mean that neutral stratification is assumed? (this might be essential when explaining the substantial differences in wind speed between observations and model)

P6, l. 12 this correction is quite unclear: what is 'corrected'? the measured wind speed? The modelled wind speed? In other words: is the modelled wind speed 'extrapolated' to the height of the observation (if so, using which input (friction velocity and

roughness length from observations? Or from the model?) or are the measured wind speeds 'corrected' to the first model level?

P6, l. 16 included into what?

P7, l. 29 'we observe...' seems to be somewhat misleading (I assume it should mean that the lowest observation level is at 2800 m). Please reformulate.

P8, l.22 'the domain...covers 30 km': with a radius of 30 km? or a radial extent? Anyway, the domain should probably be characterized by an area information not a distance.

P15, l. 2 i.e. a high median is present/ is modelled/ occurs if the large-scale....

P16, l. 22 in-depth Tab 2 caption refers to 'WRF snow precipitation' while the entries are either 'WRF 2830 m asl' or WRF total ground precip. Please clarify.

P17, l. 14 reproducing the trend: what is now the trend? Do you mean what before was called the inclination? Please clarify.

Fig. 6 If the normalized variance is shown (as indicated in the caption): why then can it be at times larger than 1? Please clarify (normalized by what). Same in Fig. 9

P20, l. 16 represent

P21, l.15 geostatistical analysis: either 'analyses show' or 'analysis shows'. However, it is not clear, whether the authors refer to their own analyses here (as presented in the foregoing sections) or to other analyses (by others), in which case a reference would be needed.

References Jimenez, P. A., and J. Dudhia, 2012: Improving the representation of resolved and unresolved topographic effects on surface wind in the WRF model. J. Appl. Meteor. & Climatol., 47, 308–325

---

## Referee Comment (RC3) · Anonymous Referee #3 · 8 May 2018

This study uses COSMO-WRF simulations to investigate the relative importance of cloud processes and the interaction between near surface flow and particles on snow accumulation in complex terrain. It shows that the WRF LES simulation, even at 50 m resolution, does not capture the particle-flow interaction near the surface in complex terrain. The authors conducted a scale analysis using autocorrelation maps and variograms to show a correlation with the terrain height, the orientation of the valleys and the precipitation distribution simulated. They also have access to radar measurements located at higher altitude to compare with the model simulations. This is a valuable study on precipitation distribution and evolution in complex terrain. I recommend the manuscript to be published after addressing these minor comments. The English

should also be improved before publication of the manuscript.

These are the specific comments:

1. The organization of the introduction should be improved. The goal of the study is not clearly stated. There is a question at the end of p.2. The following paragraph continues with the literature review and then, they elaborate a little more on the objective of the study. I would suggest focusing first on the literature review, listing the gaps in the field of snow redistribution near the surface, the objective of the study and how it will be addressed.

2. P. 4, line 1, do you mean northerly? The sub-domain is located on south-easterly of the domain.

3. P. 4, line 5, should probably give the precise dates here.

4. P. 5, line 10, the units should not be in italic. Please modify here and everywhere in the manuscript.

5. P.7, line 4: Should it be at noon instead of "during"? I am not sure what the authors mean here.

6. P.10, Figure 2: The font on the figures is too small. Letters should be added to all panels to help following the text.

7. P.11, Figure 3: The font is too small. It is very hard to read.

8. P. 11, line 8 and 9: Please clarify the sentence: "The introduction of drops in relative humidity by WRF may be due to an overestimation of subsidence in the model." Subsidence is not necessarily directly linked to relative humidity. It contributes to warm the air adiabatically, which leads to drier conditions.

9. P.12, line 12: Similar comment as #7.

10. P.13, line 12: What do you mean by "advection in the microphysics scheme"?

The microphysics scheme predicts the mixing ratio and the number concentration of hydrometeor through phase changes, collection and fall speed. I think that advection of hydrometeor is computed in WRF. Please clarify this sentence.

11. P.14, Figure 4: The font is too small. It is very hard to read. It is also hard to compare among each other because the scale is different. Please use the same scale, if possible.

12. P.15, Figure 5: The lines associated with the figure axis are missing on the figure. The x-axis could be called the Data sources (or something like that). I think that the dates could be at the top of each panel. Also please add letters to panels for clarity.

13. P.16, line 3: The following sentence states that the model is overestimating precipitation: "...model tends to overestimate precipitation for higher resolution...". As mentioned in line 21, there is also uncertainty in the radar derived precipitation amounts. It would be good to add 1-2 sentences about his issue. For example, did you try other S-R relationship to derive precipitation information from the radar?

---

## Author Comment (AC1) · 15 Jun 2018

**Response to Heini Wernli**

**'Spatial variability of snow precipitation and accumulation in COSMO-WRF simulation and radar estimations over complex terrain'**

F. Gerber (gerberf@slf.ch), N. Besic, V. Sharma, R. Mott, M. Daniels, M. Gabella, A. Berne, U. Germann, M. Lehning

Dear Heini Wernli,
We are thankful for your positive comments and for the feedback, which helped us to improve the manuscript. Reviewer comments are printed in black. Our answers and comments are printed in blue. Main changes done in the manuscript are printed in italic. **All pages and line numbers refer to the track-changes version of the manuscript (attached below).**

**Major comments ————**

A) I found it rather difficult to read the paper and follow the story-line. I very strongly recommend that the authors add a paragraph at the end of the introduction, where they clearly outline the specific objectives of the study and why they need the tools they use to address these objectives. In the current manuscript I was missing a clear roadmap: what are the objectives and questions? Why do we look at a certain diagnostic? What in the end did we learn from a specific section?

We agree that a paragraph introducing the story-line at the end of Section 1 is missing. We added an extended description which guides the reader through the different Sections and gives information on their objectives (P. 4, lines 14-30).

B) To further increase the readability, the text should be improved and several unclear statements amended. Several examples are given below in the minor comments. In several paragraphs I was left with a rather unclear impression of what I should have learned from reading the paragraph - a more reader-friendly writing style would help.

To make the text clearer we tried to simplify complex sentences. Additionally, where it was missing, we added a short paragraph or a one-sentence statement at the end of the subsections in the section Results and Discussion to assist the reader with filtering out the important statements.

**Minor comments ————**

1) p. 1 line 3: the term "particle-flow interaction" appears several times in the paper but is never explained. I am not sure whether I understand the term, is it about how the turbulent flow influences the pathway of individual snow particles? Is this really the topic of this study? Please clarify.

Yes, particle-flow interaction refers to how the local flow field (which includes turbulence and the mean wind) affects the distribution of snow particles in the air, i.e. how the local flow field affects the pathways of snow particles and thus particle distribution in the air.
We agree that this is not the main topic in this paper. However, it is one of the reasons, why very high resolution numerical simulations are required to properly simulate snow precipitation distribution in complex alpine terrain.

P. 1, line 4-6: As particle-flow interactions are not the main topic of this paper we tried to generalize this sentence: "*However, it is still challenging to quantitatively forecast precipitation especially over complex terrain, where the interaction between local wind and precipitation fields strongly affects snow distribution at the mountain ridge scale. Therefore, it is essential to retrieve high-resolution information about precipitation processes over complex terrain.*"

P. 2, line 30-31: Keeping "particle-flow interactions" in the paper, we further added an explanation of this term in the introduction: "*(i.e. the influence of the local flow field on the pathways of snow particles and particle distribution in the air)*"

2) p. 1 line 5: it looks as if COSMO-WRF was the name of the reanalysis data, but this is not what you mean - please clarify.

We changed the sentence to (P. 1, l. 6-8): "*Here, we present very high resolution Weather Research and Forecasting model (WRF) simulations (COSMO-WRF), which are initialized by 2.2 km resolution Consortium for Small-scale Modeling (COSMO) analysis.*"

3) p. 1 line 10: not clear what "variability relative to the domain wide variability" means

Changed to "*spatial variability normalized by the domain-wide variability*" (P. 1, l. 14-15)

4) p. 1 line 11: "quite substantial" is not scientific language, can you be more specific?

We changed "quite substantial" to "*systematically shows lower*" (P. 1, l. 16).

5) p. 1 line 11: The last sentence of the abstract "However, ..." seems rather unconnected to the previous sentences

We agree that this sentence is standing alone. We tried to clarify and to establish the connection by (P. 1, l. 17-P.2, l. 2):

"*A comparison of variability for different model resolutions gives evidence for an improved representation of local precipitation processes at a resolution of 50 m compared to 450 m. Additionally, differences of precipitation between 2830 m above sea level and the ground indicate that near-surface precipitation processes are active in the model.*"

6) p. 2 line 7: here and in many other places: why are references not in chronological order?

This was not on purpose. We corrected this and put all references in chronological order.

7) p. 2 line 31 and p. 5 line 20ff: you refer to the COSMO-2 analysis fields as "reanalyses", which is most likely wrong. If I am not mistaken, then the fields you use are the operationally produced hourly analysis fields. The term REanalysis would only be appropriate if the older analyses were recalculated with a newer data assimilation approach. If this is the case, then it should be explained.

This is correct; COSMO-2 was mistakenly referred to as reanalysis and we corrected this throughout the manuscript.

8) p. 3 line 1: I know what you want to say, but this is a strange sentence. Of course, relevant measurements can also be made with surface barometers, it just depends on what you want to measure!

We tried to clarify the sentence by clearly stating that we want to measure high-resolution properties of the atmosphere at different elevations above the atmosphere. The new sentence reads (P. 3, l. 25-27):

"*Remote sensing techniques, on the other hand, are the most important methods to obtain high-resolution spatial measurements of atmospheric properties at different atmospheric levels.*"

9) p. 3 line 4: "orographic mechanisms": not clear to me what you mean

With "orographic mechanisms" mechanisms leading to orographic precipitation enhancement are meant. We changed this sentence to "*These properties have been used to infer orographic precipitation enhancement, particularly ...*" (P. 3, l. 30-31).

10) p. 3 line 20: here and in other places, use km for kilometer and m for meter

Done.

11) p. 3 line 33: mb should read hPa

Done.

12) p. 4 line 2: I would rather say that this is the northeastern part of Graubünden (see also p. 6 line 19); no need to write "(CH)"

We agree that "northwestern part" is not appropriate. However, instead of calling it "northeastern" we decided to rather call it "*central northern part of the Grisons*". We kept the English version of "Graubünden" but removed "(CH)" (P. 5, l. 13-14 and P. 8, l. 24-25).

13) p. 4 line 7: I find "mesoscale mode" a bit confusing, why not just LES and non-LES mode? And - see Table 1 - is there really no PBL scheme used in LES mode? I thought that for the small-scale turbulence you still need a parameterization?

In WRF the planetary boundary layer parameterization and the subgrid-scale turbulence parameterization are two separate schemes. Indeed, there is no PBL scheme applied for LES simulations but of course there is a subgrid scale parameterization of turbulence. To clarify this in Table 1 we added a column for the subgrid scale turbulence parameterization.

Concerning the naming of the different modes – the use of "mesoscale mode" instead of non-LES mode comes from Talbot et al., 2012. However, we agree that it would be more straight forward to call the different modes LES and non-LES mode, which we changed in the whole manuscript.

14) p. 5 line 3: what is a "combined simulation"?

Combined simulation here means that the parent domain and all nests are simulated together, compared to simulations by Talbot et al., 2012 who have run a simulation with the non-LES domains and used output of the innermost non-LES domain to feed a second simulation run of the LES domains.

We tried to clarify (P. 5, l. 27-29): "*Running a nested simulation of the non-LES and LES domains turned out to be necessary for precipitation to evolve properly in the LES domains, as hydrometeors cannot be used as a boundary condition for the parent domain but are fed to nested domains in WRF simulations.*"

15) p. 6 line 10: units (here K m-1) should not be in italics (see also several other places)

Done.

16) p. 7 line 31: should read "restrict ... to only four ..."

Done.

17) p. 8 line 21: I don't understand: "three different domains" but then four resolutions?

We agree that this is confusing. The reason is that we analyze 450 m resolution precipitation patterns and 300 m resolution precipitation patterns on the regional and local domain, respectively. To compare the performance of the model simulations for different resolutions, we compare spatial variability at the different model resolutions (450 m, 150 m and 50 m) in a small domain restricted by the extent of simulations available at a resolution of 50 m (domain Dischma). This gives three domains but four resolutions.

We tried to clarify by reducing this sentence to (P. 10, l. 18-20): "*Given the resolution restriction by the radar measurements (Sect. 2.3) we analyze two different domains using horizontal resolutions of 450 m and 300 m, respectively.*"

18) p. 8 line 27: please explain what a "lag distance bin" is and for what purpose you will use the semivariance

We tried to clarify what "lag distance bin" means by "lag distance bins (h, i.e. a set of lag distance ranges). Semivariance is used to produce the variograms, whose purpose is explained at the beginning of section 2.4.

We tried to clarify by (P. 10, l. 26-27): "*To produce variograms the semivariance (γ) is calculated at 50 logarithmic lag distance bins (h, i.e. a set of lag distance ranges) by:*".

19) p. 10 last line: "For all stations ...": there are only two –> "For both stations ..."; then "matches well", hm, Fig. 2 shows deviations up to 6K, which is a lot!

We change "For all stations… " to "*For both stations…*" and weaken the statement about the matching to "*reasonably*" instead of "well" (P. 12, l. 2-P.13 l.1). Although there are strong deviations of up to 6 K we think that the matching is reasonable given that station measurements are known to show errors during midday (due to strong radiation impacts).

20) Figure 2: I find the overestimation of winds in WRF for the first two cases rather dramatic (my impression is that this should be described more clearly); is there any reason why winds are so much better for the 3rd case?

We experienced even stronger overestimations of wind speeds in different WRF simulation setups. The presented simulations are currently the best performing simulations we could achieve with our simulation setup. However, we think that WRF has a strong potential to produce improved wind speeds. We found the subgrid scale turbulence closure to have a rather strong effect on wind speeds. Unfortunately, first tests using improved subgrid scale turbulence closure schemes, which should be even better suited for complex terrain (e.g. Mirocha 2010, 2014), produced numerical instabilities in our current simulation setup. However, increasing surface roughness could partially reduce wind speeds in our model simulations.

We agree that the overestimation at station Dischma Moraine (Fig 2) is rather dramatic. The two stations are on the windward slope for the two events on 31 January 2016 and 4 February 2016, while they are on the leeward slope for the event on 5 March 2016. Therefore, it seems that the model performs better in representing the leeside flow field, while it struggles to properly reproduce the flow field on the windward side of a slope. Additionally, very local terrain features (not resolved in 50 m resolution) may have a strong influence on the local wind speed and may lead to local speed-up effects or a local reduction of wind speeds. However, reasons for the wind speed overestimation may be manifold and lack a full explanation.

Based on this and the comments from reviewer 2 we reformulated the section about wind speeds (P.14, l. 15 – P. 15, l.26):

[revised manuscript text omitted]

21) p. 11 line 5: here I find the "COSMO blaming" a bit too strong, because for instance Fig. 2 shows that COSMO is doing better for the 2nd case. Rather often, the 2-m RH values are 100%, which indicates fog. Could it be that correctly simulating fog is a particular challenge for WRF?

We agree that the COSMO blaming may be too strong in this case and we changed "is likely induced by COSMO input" to "might be induced by COSMO input". It may be a particular problem in our WRF setup to properly reproduce humidity. One reason may be that parameterizations in the model were

originally introduced for coarser resolution simulations, which experience much weaker up- and downdrafts. Due to additionally resolved topography in our model setup strong up- and downdrafts may therefore lead to an overestimation of humidity generation and decay.

Additionally, the sensors are prone to measurement errors, which may reach up to about 10% for relative humidity based on a comparison of different sensors on the Versuchsfeld on Weissfluhjoch. We tried to clarify this in the text (P.14, l. 3-6):

"*Thus, the introduction of a higher variability in relative humidity in our WRF simulations may be due to strong subsidence and lifting, which lead to an overestimation of adiabatic cooling or warming and hence to an overestimation of humidity generation and decay. Additionally, differences between modeled and measured relative humidity may be due to measurement uncertainties.*"

22) p. 12 line 5: "For some stations ...", but you only discuss two!

We changed "For some stations…" to "*For some cases…*" as we compare 3 events for two stations (e.g. P. 14, l. 13).

23) p. 12 line 10: "could partially reduce overestimated wind ...": unclear, reduce compared to what?

This is a reduction compared to WRF simulations with standard WRF roughness length for snow (0.002 m). We clarified in the sentence (P. 14, l. 19-21): "*The use of a snow roughness length of 0.2 m, … , compared to simulations with standard WRF roughness length of snow of 0.002 m, could partially reduce overestimated wind speeds (Gerber and Sharma, 2018).*"

24) p. 12 lines 12 and 20: "reasons ... may be manifold" is not a strong statement (and it should not be repeated). It seems that the "reasons are unknown", which is fine but it should be stated maybe in this way.

We changed "reasons may be manifold" to "*actual reasons for deviations in wind speed remain unknown*" (P. 15, l.23-24).

25) p 12 line 18: but if very small-scale topography matters so much, why then is COSMO performing better for some cases?

Local topographic features may lead to an enhancement or reduction of wind speeds compared to expected wind speeds for "non-disturbed" topography at a given location. For example in a valley, winds are generally channeled. If a valley is better represented in the WRF simulation and maybe even erased for a coarser model resolution, we expect WRF to simulate stronger wind speeds. However, if upwind of the station some terrain features exist, they may disturb the local flow field and measured wind speeds at the station can be rather low. In this case, it may happen that COSMO shows the better performance to simulate the actual wind speed compared to WRF but for the wrong reason.

[Figure]

However, there are most likely unknown additional reasons for the failure of WRF wind speed representation. No reason was found to explain why wind speeds are not systematically improving for higher resolution, which may be an indication that further mechanisms are essential for the wind speed representation.

26) p. 12 line 25: improve writing "in precipitation is observed for precipitation ..."

We reworded to "*The precipitation field on 31 January 2016 shows a strong south-north gradient (Fig. 4a), while the precipitation field on 4 February 2016 shows a more homogeneous distribution (Fig. 4b).*" (P. 15, l. 31-33)

27) p. 12 line 27: The sentence "Although ..." is too long and very difficult to understand, in particular, why does the sentence start with "Although"?

We used "although" because we wanted to stress, that although the domain we analyze is pretty small and within the alpine mountain range (i.e. is not extended to the northern or southern boundary of the alpine mountain range), we can still observe a precipitation gradient which is in agreement with the gradient expected for large-scale orographic precipitation enhancement. We have split the sentence in two sentences, reading (P. 15, l. 34-P. 16, l. 3):

"*Although our regional domain does not represent a cross section across the whole alpine mountain range, a north-south (south-north) precipitation gradient for southerly (northerly) advection are apparent. This is in good agreement with large-scale orographic precipitation enhancement, …*"

28) p. 12 lines 28 and 34: I am a bit lost, aren't these two sentences contradicting? Once southerly flow leads to a N-S gradient and then for southerly advection there is a S-N gradient ...

There was a mistake in the text. The S-N gradient in line 34 should be a N-S gradient as in line 28. We corrected this mistake and thank you for pointing this out. (P. 16, l.9 in track-changes version of the manuscript.)

29) p. 13 line 4: Another example where reading/understanding is difficult: "Disagreement of patterns ...": you probably mean "between WRF and radar", and then "strong smoothing of topography": you probably mean "in the model". Please improve the writing to make this type of sentences much clearer.

We made this sentence clearer by writing (P. 16, l. 16-17): *"Disagreement between radar and WRF precipitation patterns may further be connected to the strong terrain smoothing in the model."*

We additionally tried to improve such sentences throughout the manuscript (e.g. P. 16, l. 26-27 and l. 28-30).

30) p. 13 line 7: "precipitation fed to domain d02" but before, on p. 5 line 4, you wrote that "hydrometeors cannot be used as a boundary condition in WRF simulations": how does this go together? What do you mean by "feeding precipitation"?

WRF takes initial and boundary conditions for the parent domain (d01). Hydrometeors cannot be used as boundary condition in the model, but it is possible to initialize WRF with hydrometeors.

For nested domains the situation is however different. Nested domains are only initialized at the first time step by COSMO-2 in our case. Boundary conditions are taken from their parent domain and all available parameters from the parent domain are fed to the nested domain. We tried to make this clearer in Section 2.1 (P. 5, l. 27-29):

*"Running a nested simulation of the non-LES and LES domains turned out to be necessary for precipitation to evolve properly in the LES domains, as hydrometeors cannot be used as a boundary condition for the parent domain but are fed to nested domains."*

31) p. 13 line 11: you show the overestimation of winds only at 4-5m! It is not evident that winds are also too strong higher above.

This is true and unfortunately we do not have upper atmosphere wind speed information over the area of interest. We changed the sentence to a hypothetical sentence to illustrate that if wind speeds are overestimated at upper levels, advection might further strengthen the observed effect.

P. 16, l. 23-25: *"Additionally, if the tendency of overestimated wind speeds sustains up to higher atmospheric levels in the model, this may lead to an overestimation of the advection of hydrometeors in the microphysics scheme (Morrison et al., 2005). This would result in a downstream shift of the precipitation maximum."*

32) p. 13 line 19: I don't understand partially filtered vs. median-filtered

To eliminate residual clutter in the radar estimated we run a 3x3 median-filter (i.e. assigning the median of a 3x3 matrix surrounding a certain grid point to the grid point itself). Partially filtered means that the value of the median-filter is assigned to all grid cells, for which the actual values deviates more than 5 mm/h from the median-filtered value. Median-filtered means that the whole precipitation field is based on median-filtered values.

The procedure is explained in the methods part in Section 2.3 (Operational weather radar data). We added a cross-reference to Section 2.3 in the text (P. 16, l. 33).

33) Fig. 4: I find it unfortunate that the different panels for the same case have different scales. Also, I find it difficult to interpret the topography contours: for nonlocals it is not clear where valley and mountains are. A color plot of topography only could help.

To make it more intuitive to compare the patterns, we decided to show the deviation from mean precipitation, which we can show with the same colorbars. We hope this improves the readability of the figure. Additionally, we increased the font size. Using the same colorbar for all panels showing precipitation distributions was not possible due to strongly differing precipitation distributions.

We further added a shaded topography in panels k-m to help the reader understand the topography in the domain.

34) Figure 5: I don't understand the difference between the 300 and 450m resolutions: why is this so important? why has it such a large effect? and why is this effect different for radar than WRF for the first case?

The difference between 450 m and 300 m is the resolution, i.e. the Figure shows the comparison of radar precipitation, WRF precipitation @2830 m asl and WRF precipitation on the ground for the regional domain (450 m) resolution and the local domain (300 m resolution) and two different terrain smoothing strengths in the model.

The effect is so large, because the domains are different. The local domain only covers a small part of the regional domain (see Figure 1 in the manuscript). Therefore, the statistics on the local domain strongly differ depending if precipitation is weak or strong over this domain.

We agree that it may be confusing to show the statistics of two different domains in one figure. Therefore, we decided to split Figure 5 in two figures. The new Figure 5 shows only statistics for the regional domain (450 m resolution). Additionally, we add a figure to the Supplementary information, which shows statistics for the local domain (300 m resolution).

We changed the text accordingly (P. 18, l. 16-25).

35) p. 15 line 2: Hm, isn't this a complicated way of saying that the median is higher if there is more precipitation?

Yes, this sentence says that the median is higher if there is more precipitation. It was formulated that way because of the different domains covered by the 450 m resolution and 300 m resolution. However, given that we decided to split Figure 5 in two figures (see answer above), we adapted the paragraph and hopefully were able to rephrase it in a more reader-friendly way.

P. 18, l. 21-22: "*The median of radar precipitation in the local domain can be both, higher or lower than in the regional domain.*"

36) p. 15 line 3: what is meant by "shifted"? In WRF relative to radar?

Yes, this means that the pattern is shifted with respect to the precipitation pattern estimated from radar data, i.e. the radar shows more precipitation in the north-western part of the domain. We clarified the sentence by (P. 18, l. 26-31):

*"For the precipitation events on 31 January 2016 and 4 February 2016 median precipitation at 2830 m asl in the COSMO-WRF simulations is in reasonable agreement with radar median precipitation (Fig. 5), even though WRF and radar precipitation patterns are different (Section 3.2). However, for the precipitation event on 5 March 2016 the median of precipitation in the regional domain is higher in WRF simulations compared to radar measurements, while the large-scale precipitation gradient is in good agreement (Fig. 4c and Fig. 4f)."*

37) p. 16 line 15: I don't understand "strong smoothing may not allow for precipitation to evolve"

Strong smoothing leads to smaller elevation differences. For mountain peaks with small elevation gradients, this may lead to elevation differences, which are too small for efficient precipitation production. We tried to clarify this sentence by explicitly mentioning how strong smoothing may affect precipitation evolution, which now reads (P. 20, l. 12-14):

*"This tendency seems to only apply for the highest elevations. For lower elevations strong smoothing may result in elevation differences, which are too small for precipitation to evolve by lifting condensation (Sect. 3.2)."*

38) p. 17 line 21 (and in other places): not sure that "idealistic" is the right term, do you mean "simplistic"?

We agree that *"simplistic"* may be more appropriate here. We refer to the too strongly simplified representation of precipitation in the model. We changed this throughout the manuscript.

39) p. 17 line 22: what is "particle dynamics"?

What we refer to is *"cloud dynamics"*, which has mistakenly been named "particle dynamics" (P. 21, l. 28).

40) Figure 6: I first overlooked the small $10^4$ beneath the x-axis and wondered about the values. Why not just write ticks at 5, 10, ... km?

There is no particular reason for this notation. We agree that distances in km are more appropriate in this case. We changed this for Figures 6, 7 and 9 as well as for Figures S2 and S3 (previously S1 and S2). Additionally, we recognized that mistakenly the normalized semivariance was specified as $m^2$ although normalized semivariance is dimensionless. We corrected the figures for this mistake.

41) p. 22: to me, these two statements are contradicting each other: first you say that topographic features are underrepresented (I agree) and then that subsidence and lifting are too strong - but aren't they linked to topography?

Here, we mainly refer to the underrepresentation of small-scale terrain features which are likely to influence local wind speed. For lifting and subsidence, however, larger scale topography is most likely more important as vertical winds are more strongly dependent on the general flow field. Small-scale terrain features are still underrepresented in the model. The high resolution topography results in a much better representation of the general topography compared to "state-of-the-art" applications of the WRF model. Therefore, it is possible that due to the improved representation of topography, subsidence and lifting are such that they produce too strong effects in the (partially parameterized) cloud dynamics.

We tried to clarify (P. 28 l. 5-8): *"… but tends to overestimate near-surface wind speeds, which may be due to many reasons from an overestimation of speed-up effects to an underrepresentation of small terrain features. Relative humidity patterns are highly variable and may be a sign that subsidence and lifting produce too strong effects in the (partially parameterized) cloud dynamics, given the good representation of topography at larger scales."*

42) p. 23 line 17: not sure that I understand "When observing with a critical eye", what is observed here?

In this case "observing" should probably rather be replaced by "judging". What we want to express here, is that even though radar data cannot be considered as the absolute reference, it is still valuable to improve process understanding and to evaluate model simulations. However, it needs to be judged critically and it is important to keep in mind that it is not an absolute reference.

We reformulate the sentence to (P. 28, l. 26-28): *"In case of critical data analysis, an estimation of high-resolution radar precipitation is, however, useful to improve the understanding of precipitation processes in complex terrain and to validate and improve model simulations."*

43) p. 24 line 2: I don't understand this important sentence: I somehow understand that you say that WRF has too simplified cloud microphysics, but then why "due to the given resolution"? Do you say that the microphysics is OK for another resolution?

We agree that microphysics wouldn't be better represented at an even higher resolution. However, local effects of cloud dynamics might be improved if additional topography (especially less smoothing and steeper slopes) could be applied in the model. We agree that the sentence is misleading and we tried to clarify (P. 28, l.29-34):

[revised manuscript text omitted]

**S1. Model levels**

Table S.1 gives the first 21 model levels all four model domains (d01–d04).

**Table S.1.** First 21 model levels for all four model domains (d01–d04).

| | Domains | | | |
|---|---|---|---|---|
| Level | d01 | d02 | d03 | d04 |
| | m | m | m | m |
| 1 | 24.1 | 23.1 | 22.6 | 3.1 |
| 2 | 83.0 | 79.7 | 77.6 | 9.4 |
| 3 | 163.4 | 156.9 | 152.7 | 18.8 |
| 4 | 265.7 | 255.4 | 248.1 | 31.4 |
| 5 | 394.5 | 379.1 | 368.0 | 44.0 |
| 6 | 558.2 | 536.2 | 519.9 | 56.6 |
| 7 | 758.7 | 728.0 | 705.4 | 94.5 |
| 8 | 944.0 | 904.9 | 848.5 | 158.0 |
| 9 | 1098.5 | 1051.7 | 933.9 | 221.9 |
| 10 | 1255.8 | 1200.7 | 1020.1 | 286.2 |
| 11 | 1416.0 | 1352.0 | 1107.1 | 351.0 |
| 12 | 1657.3 | 1578.9 | 1237.5 | 416.2 |
| 13 | 1979.9 | 1881.7 | 1411.3 | 481.6 |
| 14 | 2305.8 | 2187.1 | 1585.9 | 547.8 |
| 15 | 2635.0 | 2495.1 | 1761.4 | 614.3 |
| 16 | 2967.6 | 2805.8 | 1937.8 | 681.2 |
| 17 | 3303.8 | 3119.3 | 2115.2 | 748.6 |
| 18 | 3643.7 | 3435.7 | 2293.7 | 811.9 |
| 19 | 3987.4 | 3755.2 | 2473.2 | 871.2 |
| 20 | 4335.0 | 4078.0 | 2653.9 | 930.8 |
| 21 | 4686.8 | 4404.4 | 2835.6 | 990.7 |

**S2. Variability at the local domain**

Figure S.1 show the domain-wide statistics of the local domain, for which data has a resolution of 300 m (Figure 1 in the main text). Variograms analogously to Figure 6 and Figure 7 in the main text are presented for the local domain. Trends removed from the data to produce the variograms in Figure S.2 are given in Table S.2. For Figure S.3 no trends are removed to retain the effect of large-scale precipitation processes. Thus, small as well as intermediate scale patterns may be hidden by the domain-wide trends.

**Table S.2.** Large-scale linear trends of entirely-filtered radar and WRF precipitation patterns on the local domain (Figure 1 in the main text). *Orientation* gives the direction of the slope and *Intensity* the strength of inclination. 0° would indicate a slope pointing toward the East. WRF snow precipitation is from simulations with weak terrain smoothing (Sect. 2.1 in the main text).

| | 31 January 2016 | | 4 February 2016 | | 5 March 2016 | |
|---|---|---|---|---|---|---|
| | Orientation | Intensity | Orientation | Intensity | Orientation | Intensity |
| Radar entirely filtered | 68.5° | 0.14 | 150.5° | 0.03 | -135.4° | 0.04 |
| WRF precip. at 2830 m asl | 22.6° | 0.26 | 5.8° | 0.24 | -79.6° | 0.19 |
| WRF total ground precip. | 30.6° | 0.32 | 24.2° | 0.24 | -67.6° | 0.21 |

[Figure]

**Figure S.1.** Domain-wide 24 h precipitation statistics for the local domain (300 m resolution, Figure 1 in the main text) for the three precipitation events on 31 January 2016, 4 February 2016 and 5 March 2016. Gray colors show entirely-filtered radar precipitation. WRF precipitation at 2830 m above sea level (m asl) for simulations with weak terrain smoothing (Sect. 2.1 in the main text) and strong terrain smoothing are given in blue and violet, respectively. Orange (red) shows boxplots of WRF total ground precipitation for weak (strong) terrain smoothing. Radar precipitation and WRF precipitation at 2830 m asl are masked (as shown in Figure 4 in the main text).

[Figure]

**Figure S.2.** Normalized variograms of detrended snow precipitation for the precipitation events on a) 31 January 2016, b) 4 February 2016 and c) 5 March 2016 for the local domain (Figure 1 in the main text). Variograms are given for partly-filtered (red) and entirely-filtered (orange) radar snow precipitation, WRF snow precipitation at 2830 m above sea level (m asl, blue) and WRF total ground precipitation (violet). WRF precipitation is from simulations with weak terrain smoothing (Sect. 2.1 in the main text). All precipitation fields are masked.

[Figure]

**Figure S.3.** Normalized variograms of the snow precipitation events on a) 31 January 2016, b) 4 February 2016 and c) 5 March 2016 for the local domain (Figure 1 in the main text). Variograms are given for entirely-filtered radar snow precipitation (orange), WRF snow precipitation at 2830 m above sea level (m asl, blue) and WRF total ground precipitation (violet). Additionally, variograms are given for real topography (based on dhm25 © 2018 swisstopo (5740 000 000), black) and WRF topography (gray). WRF topography and precipitation are from simulations with weak terrain smoothing (Sect. 2.1 in the main text). All precipitation fields are masked.

---

## Author Comment (AC2) · 15 Jun 2018

**Response to Reviewer #2**

**'Spatial variability of snow precipitation and accumulation in COSMO-WRF simulation and radar estimations over complex terrain'**

F. Gerber (gerberf@slf.ch), N. Besic, V. Sharma, R. Mott, M. Daniels, M. Gabella, A. Berne, U. Germann, M. Lehning

Dear Sir or Madam,
We are thankful for your positive comments and for the feedback, which helped us to improve the manuscript. Reviewer comments are printed in black. Our answers and comments are printed in blue. Main changes done in the manuscript are printed in italic. **All pages and line numbers refer to the track-changes version of the manuscript (attached below).**

**Major comments ————**

1) Explanation of WRF offsets (Section 3.1, point inter-comparison). It is argued that these offsets were (partly) 'due to offsets in the COSMO-2 input' (p11, l. 2/5; p12, l. 11). This is a weak argument. Either the domain and time settings (e.g., distance from the inflow domain border; spin-up time) are chosen in a way that allows the downscaled model run to develop its own local conditions - which would demonstrate the improvement (or deterioration) due to the higher resolution, or the downscaling is essentially useless. This argument should be re-considered. Furthermore, for wind speed in particular, e.g. Jimenez and Dudhia (2012) come to a quite different conclusion than the present authors (but based on simulations for which the roughness length is not artificially enhanced). The present results should be at least discussed in the light of their results. Also, it should be noted that apparently WRF can produce some 8 m/s wind speed at times, while the observation is around 2 m/s or even less. On other days (different wind direction!) the WRF bias is not nearly as large. This does not seem to be a problem of mere roughness length or some problem in the 'model physics'. To me, this rather looks like a problem with the height attribution (or 'correction' - using a logarithmic [neutral?] profile - of the modelled profile to the observation height). Could it be that the actual observation height was '2 m minus snow depth' (rather than 2 m as in the model)? So, in case of a deep snow layer at the observational site the actual observation height would possibly be much smaller. In any case, this issue should be analysed in some more detail.

We agree that the overestimation in COSMO-2 input may not be the main argument why wind speeds are overestimated. However, if the model is constantly fed by excess wind speeds it may not be able to correct for this overestimation. Accordingly, we weakened the argument of COSMO-2 input being the main reason for overestimated wind.

Indeed Jimenez and Dudhia (2012, JD12 hereafter) used a quite different approach to improve simulated wind speeds in WRF simulations. However, there are some differences between the simulations by JD12 and our simulations, which led us to stick to a simple correction of the roughness length. First of all, the finest model resolution in JD12 is 2 km, while the coarsest domain in our simulation setup already has a higher resolution of 1.35 km. We only use a planetary boundary layer (PBL) parameterization on domain d01, while domains d02-d04 (with

resolutions as fine as 50 m) are run in the large-eddy simulation (LES) mode. However the subgrid-scale topography correction is implemented in the PBL scheme and would therefore only affect domain d01 in our simulation setup. Additionally, simulations by JD12 are performed over snow-free complex terrain, while our simulations are performed over entirely snow covered terrain. This means that the standard roughness length supplied by WRF for our simulation approach would be 0.002 m (the value for snow). At the scale of our high resolution simulations, we argue that complex terrain features (e.g. boulders and rocky outcroppings) affect the average roughness length, which in itself is a theoretical value to allow a modeled approximation of boundary layer flow. Thus, overall we think that using a model approach with an increased roughness length is justified for our simulation setup at very high resolutions (LES-mode) in complex snow covered terrain. We agree that for future work, it could be interesting and useful to perform a sensitivity analysis to additionally include the subgrid-scale topography parameterization introduced by Jimenez and Dudhia (2012) for our parent domain d01, which is run with a PBL scheme. We address these differences in the manuscript to clarify our choice. We added a discussion of this issue on P. 14, l. 24-34:

*"While we address non-resolved topography based on an increased snow surface roughness length, another approach to improve wind speeds in WRF simulations has been introduced by Jiménez and Dudhia (2012), who use a sink term in the momentum equation based on subgrid-scale topography. They demonstrate the ability of their approach to improve surface wind speeds. However, with this approach, the effect of the subgrid scale topography is only included for simulations using a PBL parameterization. As in our model setup a PBL parameterization is only applied for domain d01, we address the non-resolved topography by increasing the surface roughness, which allows us to include the effect of non-resolved topography for all four simulation domains. In addition, our simulations are run over snow covered terrain, which implies that the standard roughness length for snow used in WRF (on the order of $10^{-3}$ m) is two orders of magnitude lower than roughness lengths representative of the scale of complex terrain in our simulations (on the order of $10^{-1}$ m). Applying the PBL version of Jiménez and Dudhia (2012) might be a possibility to reduce excess wind speeds in domain d01, which might also impact wind speeds in the domains d02-d04. However, such a sensitivity study is out of scope of the present investigation. "*

Indeed wind speeds fit better for the precipitation event on 5 March 2016, while wind speed is overestimated on 31 January 2016 and 4 February 2016. We mainly attribute this to the location of the station with respect to the ridge and wind directions. Both stations are located on the northern side of ridge and therefore are on the windward side of the ridge for the two precipitation event on 31 January 2016 and 4 February 2016, while they are on the leeward side of the ridge for the precipitation event on the 5 March 2016. We therefore hypothesize that the wind field on the leeward side of the mountain ridge is better represented than the wind field on the windward side of the mountain ridge, which would be in agreement with previous studies who found speed up of wind speeds in ridge areas (e.g. Mott et al, 2010, Gomez-Navarro et al., 2015). However, you are correct that the elevation of wind speed measurements at the stations depends on the snow cover and is "SensorHeight-SnowDepth". Therefore, we recalculate wind speeds for average snow depths for the three dates and include the new figures. This results in slightly weaker wind speeds for COSMO-WRF, but it is not able to explain excess wind speed in general.

2) Overall conclusions: the authors nicely demonstrate the impact of topography (in different model resolutions) on the quality of the spatial variability of precipitation estimates by the model. In the conclusions, however, the arguments are somehow mixed up. First, (p24, l.2) it is argued that the difficulty in reproducing the small-scale variability is 'most likely [due to] the representation of cloud dynamics and microphysics in the model [being] too idealistic'. This is then supported by a number of arguments, all referring to the topography (…). Then, some few lines later (p24, l. 6) it is concluded 'This shows that especially for small-scale variability a better

representation of the complex terrain is essential to reproduce precipitation variability'. So, what is it now? The terrain representation or the microphysics being to idealized? In my view, the results of the authors have demonstrated the impact of terrain (and resolution), but no conclusions can be drawn concerning the microphysics and cloud dynamics. Certainly, the authors have not done that – and so, such a 'conclusion' concerning microphysics and cloud dynamics cannot be maintained.

3)

We agree that the formulation of the conclusion concerning reasons for missing precipitation variability is not properly chosen. It is true that we did not perform an analysis, which allows us to evaluate the representation of microphysics and cloud dynamics in the model. However, they are parameterized in the model and parameterizations always go along with simplifications. Therefore, the representation of microphysical processes is simplified and we assume this has an impact on the representation of precipitation variability in the model. On the other hand, as you point out, we investigated the impact of topography which clearly has an effect on precipitation variability but cannot explain the full lack of precipitation variability. We reformulated the conclusion concerning missing precipitation variability (P. 28, l. 18 29-34):
*"This indicates that the model is not able to represent the full spectrum of small-scale precipitation patterns, which are present in the radar measurements. One potential reason for the lack of precipitation variability is the simplification of cloud dynamics and microphysics in the model, typically used to model regional scale precipitation fields. Additionally, we could show that the underrepresentation of topography may have a strong influence on the formation of local low-level clouds, which are important for orographic precipitation enhancement."*

**Minor comments ————**

P3, l.1 means, not mean
Based on Review#1 this sentence has been changed. Means does not longer appear in the sentence.

P3, l. 33 hPa, not mb
Done.

P4, l. 4 set up, not setup
Done.

P4, l. 4 with refined vertical levels: from the above I understand that also d3 has refined vertical levels (60). Does this sentence mean, that these are not within the boundary layer? Please explain.
Yes, both domains d03 and d04 have refined vertical levels with respect to domains d01 and d02. For d03 60 vertical levels are generated based on the WRF algorithm and for d04 80 vertical levels are generated based on the WRF algorithm. This refinement affects the whole vertical extent of the atmosphere. For d04 the boundary layer is additionally refined with 10 additional model levels.
We moved the information about the additional vertical refinement to the statement about vertical nesting and tried to clarify (P. 5, l. 8-11):

*"Domains d02-d04 have 40, 60 and 90 vertical levels, respectively, with the model top at 150 hPa using a preliminary version of vertical nesting (Daniels et al., 2016). Twenty and 40 vertical levels are refining the whole atmosphere in d03 and d04, respectively. Ten vertical levels in d04 are introduced to additionally refine the boundary layer."*

P4, l.6 nests run in 'LES'-mode: for d2 (450 m horizontal grid spacing) and possibly d3 (150 m) this is in the middle of the 'Gray Zone' of Wyngaard (2004). Can the authors comment on that?

Indeed domains d02 and d03 are in the 'Gray Zone'. We ran test simulations both with d02 and d03 in the LES mode or in the non-LES mode (using a PBL parameterization). Results did not strongly differ. Therefore, and as the model setup by Talbot et al., 2012 was based on the same approach we decided to run the domains d02 and d03 in the LES mode.

An interesting approach was chosen by Muñoz-Esparza et al., 2017, who break the rule of the 1:3 grid refinement ratio to omit resolutions in the 'Gray Zone'. This approach is a very interesting approach and would be worth to be tested in future simulations over complex alpine terrain. However, as mentioned above we kept our approach close to Talbot et al., 2012 and stuck to the well tested grid ratio of 1:3 between parent and nested domains.

We added a comment on this in the paper (P.5, l.22-25): "*Domains d02 and d03 are within the 'gray zone' (Wyngaard, 2004). There are approaches omitting simulations in the 'gray zone' by the choice of a higher grid refinement ratio (Muñoz-Esparza et al., 2017), which would be worth a sensitivity study. However, we use the well-tested 1:3 grid refinement ratio and keep our model setup consistent with the very high resolution simulations by Talbot et al. (2012), …*"

Tab 1 please add the heights of the first few model levels
We decided to add the number of model levels in the lowest 1000 m of the atmosphere for the four domains in Table 1. Additionally, we added a table with the lowest 21 model levels in the Appendix.

P6, l.11 logarithmic wind profile: does this mean that neutral stratification is assumed? (this might be essential when explaining the substantial differences in wind speed between observations and model)
Yes, as the logarithmic wind profile is defined for neutral conditions, we indeed assume neutral conditions for this correction. We include this in the discussion of the excess wind speeds. However, especially for domain d04 we expect differences to have a minor impact as the lowest model level, which is used for the correction, is at a similar elevation as the station (1-2 m difference).

P. 8, l. 12-15: "*Modeled wind speeds are extrapolated to the measurement height by applying a logarithmic wind profile, as wind measurements at the automatic weather stations are not taken at 10 m but 4 or 5 m above ground (Sect. 2.2). This is a rough approximation given the assumption of a neutral atmosphere.*"

P. 15, l. 20-21: "*An additional source of uncertainty – though unlikely to be on the order of the strong excess wind speeds – is the extrapolation of wind speeds based on the assumption of a neutral atmosphere.*"

P6, l. 12 this correction is quite unclear: what is 'corrected'? the measured wind speed? The modelled wind speed? In other words: is the modelled wind speed 'extrapolated' to the height of the observation (if so, using which input (friction velocity and roughness length from observations? Or from the model?) or are the measured wind speeds 'corrected' to the first model level?
The modeled velocity is extrapolated from the nearest model level or the 10m wind speed depending on the simulation domain. For d04 modeled wind speed is extrapolated from the lowest model level (~3 m above ground) while for domains d01-d03 the 10-m wind speed is used for the correction. The roughness length is chosen to be equal to the roughness length applied in the model which is 0.2 m. The correction is based on the formula:

$$u(z) = u_r \frac{\ln(\frac{z}{z_0})}{\ln(\frac{z_r}{z_0})},$$

where z is the elevation to correct for, $z_r$ is the reference elevation and $z_0$ is the roughness length. Therefore, the friction velocity is not required.

We tried to clarify what exactly is corrected (P. 8, l. 12-17): "*Modeled wind speeds are extrapolated to the measurement height by applying a logarithmic wind profile, as wind measurements at the automatic weather stations are not taken at 10 m but 4 or 5 m above ground (Sect. 2.2). This is a rough approximation given the assumption of a neutral atmosphere. For simulation domains d01-d03 10-m wind speeds are extrapolated to the elevation of the sensor above the snow cover, while for domain d04 wind speeds at the lowest model level (approximately 3 m above ground) are used for the correction.*"

P6, l. 16 included into what?
COSMO variables are included in the model validation and therefore displayed in Figures 2 and 3. We specified this in the text (P. 8, l. 20-21):
"*... are included in the model validation and hence in Fig. 2 and Fig. 3.*"

P7, l. 29 'we observe ...' seems to be somewhat misleading (I assume it should mean that the lowest observation level is at 2800 m). Please reformulate.
Yes, this means that we use radar measurements from above 2800 m asl. We reformulated to (P. 9, l. 23):
"*... (all radar measurements are from above 2800 m asl during the winter season)*"

P8, l.22 'the domain ... covers 30 km': with a radius of 30 km? or a radial extent? Anyway, the domain should probably be characterized by an area information not a distance.
Yes, the domain covers an area centered over the radar with a radius of about 30 km. We agree that it makes more sense to precisely describe the size of the domain and changed the sentence to (P. 10, l. 20-21):
"*...covers an area of about 58 km times 56 km centered over the radar on Weissfluhgipfel...*"

P15, l. 2 i.e. a high median is present/ is modelled/ occurs if the large-scale ....

Based on comments by reviewer 1 we decided to split Figure 5 in two separate figures. We changed the text accordingly (P. 18, l. 16-25).

P16, l. 22 in-depth Tab 2 caption refers to 'WRF snow precipitation' while the entries are either 'WRF 2830 m asl' or WRF total ground precip. Please clarify.
We clarify in Table 2 (P. 22): "*WRF precip. at 2830 m asl refers to solid precipitation in WRF simulations at 2830 m above sea level and WRF total ground precip. refers to the total (solid and liquid) precipitation at the ground level.*"

P17, l. 14 reproducing the trend: what is now the trend? Do you mean what before was called the inclination? Please clarify.
When we talk about the trend, we refer to the orientation and intensity of the plane which is linearly fitted to the precipitation field. We agree that the terminology was confusing as we use different terms for the orientation (orientation and direction) of the plane as well as for the intensity of the trend (inclination and intensity). We only use orientation and intensity of the trend referring to the inclination and orientation of the linearly fitted plane, which represents the trend of the precipitation patterns. Please, check the track-changes version of the paper for all the adaptations we have done (P. 21, l. 8-31).

Fig. 6 If the normalized variance is shown (as indicated in the caption): why then can it be at times larger than 1? Please clarify (normalized by what). Same in Fig. 9

The Variograms are normalized by the domain-wide variance of the precipitation field. We clarified this in the figure captions of Figures 6 and 9.

P. 23, l. 1: *"Variograms of detrended snow precipitation normalized by the domain-wide variance of precipitation for the precipitation events…"*

The variograms are experimental. Although the general trend of the precipitation variable is subtracted from the precipitation field, the variable is not exactly stationary. Therefore, the normalized variogram may show larger variances than the domain-wide variance for certain lag distances.

P20, l. 16 represent
Done.

P21, l.15 geostatistical analysis: either 'analyses show' or 'analysis shows'. However, it is not clear, whether the authors refer to their own analyses here (as presented in the foregoing sections) or to other analyses (by others), in which case a reference would be needed.

Here, we refer to our own geostatistical analyses. We clarified by writing (P. 26, l. 1): *"
[revised manuscript text omitted]

**S1. Model levels**

Table S.1 gives the first 21 model levels all four model domains (d01–d04).

**Table S.1.** First 21 model levels for all four model domains (d01–d04).

| | Domains | | | |
|---|---|---|---|---|
| Level | d01 | d02 | d03 | d04 |
| | m | m | m | m |
| 1 | 24.1 | 23.1 | 22.6 | 3.1 |
| 2 | 83.0 | 79.7 | 77.6 | 9.4 |
| 3 | 163.4 | 156.9 | 152.7 | 18.8 |
| 4 | 265.7 | 255.4 | 248.1 | 31.4 |
| 5 | 394.5 | 379.1 | 368.0 | 44.0 |
| 6 | 558.2 | 536.2 | 519.9 | 56.6 |
| 7 | 758.7 | 728.0 | 705.4 | 94.5 |
| 8 | 944.0 | 904.9 | 848.5 | 158.0 |
| 9 | 1098.5 | 1051.7 | 933.9 | 221.9 |
| 10 | 1255.8 | 1200.7 | 1020.1 | 286.2 |
| 11 | 1416.0 | 1352.0 | 1107.1 | 351.0 |
| 12 | 1657.3 | 1578.9 | 1237.5 | 416.2 |
| 13 | 1979.9 | 1881.7 | 1411.3 | 481.6 |
| 14 | 2305.8 | 2187.1 | 1585.9 | 547.8 |
| 15 | 2635.0 | 2495.1 | 1761.4 | 614.3 |
| 16 | 2967.6 | 2805.8 | 1937.8 | 681.2 |
| 17 | 3303.8 | 3119.3 | 2115.2 | 748.6 |
| 18 | 3643.7 | 3435.7 | 2293.7 | 811.9 |
| 19 | 3987.4 | 3755.2 | 2473.2 | 871.2 |
| 20 | 4335.0 | 4078.0 | 2653.9 | 930.8 |
| 21 | 4686.8 | 4404.4 | 2835.6 | 990.7 |

**S2. Variability at the local domain**

Figure S.1 show the domain-wide statistics of the local domain, for which data has a resolution of 300 m (Figure 1 in the main text). Variograms analogously to Figure 6 and Figure 7 in the main text are presented for the local domain. Trends removed from the data to produce the variograms in Figure S.2 are given in Table S.2. For Figure S.3 no trends are removed to retain the effect of large-scale precipitation processes. Thus, small as well as intermediate scale patterns may be hidden by the domain-wide trends.

**Table S.2.** Large-scale linear trends of entirely-filtered radar and WRF precipitation patterns on the local domain (Figure 1 in the main text). *Orientation* gives the direction of the slope and *Intensity* the strength of inclination. $0°$ would indicate a slope pointing toward the East. WRF snow precipitation is from simulations with weak terrain smoothing (Sect. 2.1 in the main text).

| | 31 January 2016 | | 4 February 2016 | | 5 March 2016 | |
|---|---|---|---|---|---|---|
| | Orientation | Intensity | Orientation | Intensity | Orientation | Intensity |
| Radar entirely filtered | 68.5° | 0.14 | 150.5° | 0.03 | -135.4° | 0.04 |
| WRF precip. at 2830 m asl | 22.6° | 0.26 | 5.8° | 0.24 | -79.6° | 0.19 |
| WRF total ground precip. | 30.6° | 0.32 | 24.2° | 0.24 | -67.6° | 0.21 |

[Figure]

**Figure S.1.** Domain-wide 24 h precipitation statistics for the local domain (300 m resolution, Figure 1 in the main text) for the three precipitation events on 31 January 2016, 4 February 2016 and 5 March 2016. Gray colors show entirely-filtered radar precipitation. WRF precipitation at 2830 m above sea level (m asl) for simulations with weak terrain smoothing (Sect. 2.1 in the main text) and strong terrain smoothing are given in blue and violet, respectively. Orange (red) shows boxplots of WRF total ground precipitation for weak (strong) terrain smoothing. Radar precipitation and WRF precipitation at 2830 m asl are masked (as shown in Figure 4 in the main text).

[Figure]

**Figure S.2.** Normalized variograms of detrended snow precipitation for the precipitation events on a) 31 January 2016, b) 4 February 2016 and c) 5 March 2016 for the local domain (Figure 1 in the main text). Variograms are given for partly-filtered (red) and entirely-filtered (orange) radar snow precipitation, WRF snow precipitation at 2830 m above sea level (m asl, blue) and WRF total ground precipitation (violet). WRF precipitation is from simulations with weak terrain smoothing (Sect. 2.1 in the main text). All precipitation fields are masked.

[Figure]

**Figure S.3.** Normalized variograms of the snow precipitation events on a) 31 January 2016, b) 4 February 2016 and c) 5 March 2016 for the local domain (Figure 1 in the main text). Variograms are given for entirely-filtered radar snow precipitation (orange), WRF snow precipitation at 2830 m above sea level (m asl, blue) and WRF total ground precipitation (violet). Additionally, variograms are given for real topography (based on dhm25 © 2018 swisstopo (5740 000 000), black) and WRF topography (gray). WRF topography and precipitation are from simulations with weak terrain smoothing (Sect. 2.1 in the main text). All precipitation fields are masked.

---

## Author Comment (AC3) · 15 Jun 2018

**Response to Reviewer #3**

**'Spatial variability of snow precipitation and accumulation in COSMO-WRF simulation and radar estimations over complex terrain'**

F. Gerber (gerberf@slf.ch), N. Besic, V. Sharma, R. Mott, M. Daniels, M. Gabella, A. Berne, U. Germann, M. Lehning

Dear Sir or Madam,
We are thankful for your positive comments and for the feedback, which helped us to improve the manuscript. Reviewer comments are printed in black. Our answers and comments are printed in blue. Main changes done in the manuscript are printed in italic. **All pages and line numbers refer to the track-changes version of the manuscript (attached below).**

**General comment:**

The English should also be improved before publication of the manuscript.

We tried to improve the English and readability of the manuscript.

**These are the specific comments:**

1. The organization of the introduction should be improved. The goal of the study is not clearly stated. There is a question at the end of p.2. The following paragraph continues with the literature review and then, they elaborate a little more on the objective of the study. I would suggest focusing first on the literature review, listing the gaps in the field of snow redistribution near the surface, the objective of the study and how it will be addressed.

We rearranged the introduction such that it hopefully is more reader-friendly. We tried to give it a better structure and clearly highlight the research questions. At the end of the introduction we added an extended paragraph explaining the structure of the paper and related objectives (P. 2-4).

2. P. 4, line 1, do you mean northerly? The sub-domain is located on south-easterly of the domain.

Here, we only refer to the location of domains d01 and d02. We indeed mean that we shifted domain d02 toward the eastern boundary of domain d01, which is visible in the left panel of Figure 1 (d01: dark red, d02: red).

3. P. 4, line 5, should probably give the precise dates here.

We added the exact dates of the event and additionally refer to Section 2.5, where we also give information about the different snowfall event.

P. 5, l. 15-17: *"Simulations are performed for three snow precipitation events on 31 January 2016, 4 February 2016 and 5 March 2016 (Sect. 2.5)."*

4. P. 5, line 10, the units should not be in italic. Please modify here and everywhere in the manuscript.

Done.

5. P.7, line 4: Should it be at noon instead of "during"? I am not sure what the authors mean here.

We agree that "during noon" has to be changed. We change it to *"around midday"* to make sure we are talking about few hours around midday (P. 8, l. 31).

6. P.10, Figure 2: The font on the figures is too small. Letters should be added to all panels to help following the text.

We increased the font sizes in Figure 2 and added letters for each panel. Cross references in the text are adapted accordingly.

7. P.11, Figure 3: The font is too small. It is very hard to read.

As for Figure 2 (see above).

8. P. 11, line 8 and 9: Please clarify the sentence: "The introduction of drops in relative humidity by WRF may be due to an overestimation of subsidence in the model." Subsidence is not necessarily directly linked to relative humidity. It contributes to warm the air adiabatically, which leads to drier conditions.

We try to clarify by (P. 14, l.3-6): "*Thus, the introduction of a higher variability in relative humidity in our WRF simulations may be due to strong subsidence and lifting, which lead to an overestimation of adiabatic cooling or warming and hence to an overestimation of humidity generation and decay. Additionally, differences between modeled and measured relative humidity may be due to measurement uncertainties.*"

9. P.12, line 12: Similar comment as #7.

The authors are not sure what you are referring to. Comment #7 is about Figure 2 but there is no figure on P.12, line 12. Anyway the paragraph about wind speeds (which contains line 12 on P. 12) has been rewritten given reviewer comments in reviews #1 and #2.

10. P.13, line 12: What do you mean by "advection in the microphysics scheme"? The microphysics scheme predicts the mixing ratio and the number concentration of hydrometeor through phase changes, collection and fall speed. I think that advection of hydrometeor is computed in WRF. Please clarify this sentence.

The microphysics scheme by Morrison et al., 2005 "…contains kinetic equations for the mixing ratio q, and number concentration, N, …". The spatial derivatives of mixing ratio and number concentration contain an advection term (first term on the right hand side (rhs) in eqs. (1) and (2)), a sedimentation term which depends on the terminal fall velocity ($V_{qx}$, second term on the rhs in eqs. (1) and (2)) and a turbulent diffusion term (third term on the rhs in eqs. (1) and (2)). All the other terms on the rhs in eqs. (1) and (2) are source and sink terms due to microphysical processes. For more details, please, consult Morrison et al., 2005.

$$\frac{\partial q}{\partial t} = -\nabla \cdot (\mathbf{v}q) + \frac{\partial}{\partial z}(V_{qx}) + \nabla_D q + \left(\frac{\partial q}{\partial t}\right)_{\text{PRO}}$$
$$+ \left(\frac{\partial q}{\partial t}\right)_{\text{COND/DEP}} + \left(\frac{\partial q}{\partial t}\right)_{\text{AUTO}} + \left(\frac{\partial q}{\partial t}\right)_{\text{COAG}}$$
$$+ \left(\frac{\partial q}{\partial t}\right)_{\text{MLT/FRZ}} + \left(\frac{\partial q}{\partial t}\right)_{\text{MULT}}, \qquad (1)$$

$$\frac{\partial N}{\partial t} = -\nabla \cdot (\mathbf{v}N) + \frac{\partial}{\partial z}(V_{Nx}) + \nabla_D N + \left(\frac{\partial N}{\partial t}\right)_{\text{PRO}}$$
$$+ \left(\frac{\partial N}{\partial t}\right)_{\text{EVAP/SUB}} + \left(\frac{\partial N}{\partial t}\right)_{\text{AUTO}} + \left(\frac{\partial N}{\partial t}\right)_{\text{SELF}}$$
$$+ \left(\frac{\partial N}{\partial t}\right)_{\text{COAG}} + \left(\frac{\partial N}{\partial t}\right)_{\text{MLT/FRZ}} + \left(\frac{\partial N}{\partial t}\right)_{\text{MULT}}, \qquad (2)$$

11. P.14, Figure 4: The font is too small. It is very hard to read. It is also hard to compare among each other because the scale is different. Please use the same scale, if possible.

To make it more intuitive to compare the patterns, we decided to show the deviation from mean precipitation, which we can show with the same colorbars. We hope this improves the readability of the figure. Additionally, we increased the font size. Using the same colorbar for all panels showing precipitation distributions was not possible due to strongly differing precipitation distributions.

12. P.15, Figure 5: The lines associated with the figure axis are missing on the figure. The x-axis could be called the Data sources (or something like that). I think that the dates could be at the top of each panel. Also please add letters to panels for clarity.

This figure only contains one panel, thus we think that is not necessary to add panel labels. The x-axis in this figure represents time. Grouped box plots for each date are shown. The data sources are given in the legend and thus we think it is not necessary to state this more specifically. We agree that the horizontal lines associated with precipitation rates given on the y-axis were not well visible. We increase the line thickness of these lines and chose a darker color to hopefully make this figure more reader-friendly.

13. P.16, line 3: The following sentence states that the model is overestimating precipitation: "… model tends to overestimate precipitation for higher resolution…". As mentioned in line 21, there is also uncertainty in the radar derived precipitation amounts. It would be good to add 1-2 sentences about his issue. For example, did you try other S-R relationship to derive precipitation information from the radar?

Indeed, a fair point! We have not compared different S-Z relationships, but rather adopted the commonly used Finish formula. Although the uncertainty in the Z-S parameters may lead to significant uncertainty in the total amount, it should not strongly affect the spatial variability (quantified by the normalized variogram or autocorrelation). To be rigorous, we nevertheless follow the suggestion of the reviewer and explicitly mention the possible uncertainty associated with radar estimates.

P. 18, l. 22-25: *"Although radar estimates are based on a referent S-Z relationship, the employed formula (Eq. 1) is not immune to the potential estimation errors. Therefore, despite reasonably assuming that the potential estimation errors should not significantly influence the variability and the intensity of the precipitation fields, while drawing conclusions we have subjectively considered the impact of imperfections characterizing such an empirical formula."*

Reference:

[revised manuscript text omitted]

10  knoll) areWRF topography even at 50 m resolution. Such small-scale features introduce very local flow features and may either induce local speed-up effects or can reduce wind speedsdue to enhanced roughness~~the model, may strongly reduce wind speeds in reality. For station Dischma Moraine on the 31 January 2016 and the 4 February 2016 an overestimation of wind speed is already observed in COSMO–2, which might be an additional reason for wind speed overestimation at this station. However, given the fetch distances and spin-up times of our model simulations

15  (Sect. 2.1), we expect the atmosphere to develop independently. Still, if COSMO wind speeds are constantly overestimated WRF may not be able to correct for excess wind speeds. Station measurements are also prone to measurement uncertainties and riming of the unheated instruments may lead to an underestimation of wind speeds (Grünewald et al., 2012).

Generally, reasons for an overestimation of wind speeds may be manifold. An exact estimation of wind speeds at stations in the model is  not expected due to unresolved topographical features in the

20  complex terrain of our study site. An additional source of uncertainty – though unlikely to be on the order of the strong excess wind speeds – is the extrapolation of wind speeds based on the assumption of a neutral atmosphere. While different potential causes of wind speed overestimation are discussed above, actual reasons for deviations in wind speed remain unknown.

Overall, we show that the presented simulation setup reasonably captures temperature, relative humidity and wind conditions

25  in complex terrain at two stations. Wind speeds on the windward side of the mountain ridges tend to be overestimated. Temperature deviations around midday are likely due to measurement uncertainties.

**3.2 Spatial snow precipitation and accumulation patterns**

 Radar precipitation maps of the regional domain covering  an area of about 58 km times 56 km centered over the radar on Weissfluhgipfel ( Fig. 1) tend to show wind direction (~~Figure 2, Figure 3and

30  FigureFigureA strong south-north gradient in precipitation is observed for precipitation, while precipitationis located within the Alps~~
[revised manuscript text omitted]

**S1. Model levels**

Table S.1 gives the first 21 model levels all four model domains (d01–d04).

**Table S.1.** First 21 model levels for all four model domains (d01–d04).

| Level | Domains d01 m | d02 m | d03 m | d04 m |
|---|---|---|---|---|
| 1 | 24.1 | 23.1 | 22.6 | 3.1 |
| 2 | 83.0 | 79.7 | 77.6 | 9.4 |
| 3 | 163.4 | 156.9 | 152.7 | 18.8 |
| 4 | 265.7 | 255.4 | 248.1 | 31.4 |
| 5 | 394.5 | 379.1 | 368.0 | 44.0 |
| 6 | 558.2 | 536.2 | 519.9 | 56.6 |
| 7 | 758.7 | 728.0 | 705.4 | 94.5 |
| 8 | 944.0 | 904.9 | 848.5 | 158.0 |
| 9 | 1098.5 | 1051.7 | 933.9 | 221.9 |
| 10 | 1255.8 | 1200.7 | 1020.1 | 286.2 |
| 11 | 1416.0 | 1352.0 | 1107.1 | 351.0 |
| 12 | 1657.3 | 1578.9 | 1237.5 | 416.2 |
| 13 | 1979.9 | 1881.7 | 1411.3 | 481.6 |
| 14 | 2305.8 | 2187.1 | 1585.9 | 547.8 |
| 15 | 2635.0 | 2495.1 | 1761.4 | 614.3 |
| 16 | 2967.6 | 2805.8 | 1937.8 | 681.2 |
| 17 | 3303.8 | 3119.3 | 2115.2 | 748.6 |
| 18 | 3643.7 | 3435.7 | 2293.7 | 811.9 |
| 19 | 3987.4 | 3755.2 | 2473.2 | 871.2 |
| 20 | 4335.0 | 4078.0 | 2653.9 | 930.8 |
| 21 | 4686.8 | 4404.4 | 2835.6 | 990.7 |

**S2. Variability at the local domain**

Figure S.1 show the domain-wide statistics of the local domain, for which data has a resolution of 300 m (Figure 1 in the main text). Variograms analogously to Figure 6 and Figure 7 in the main text are presented for the local domain. Trends removed from the data to produce the variograms in Figure S.2 are given in Table S.2. For Figure S.3 no trends are removed to retain the effect of large-scale precipitation processes. Thus, small as well as intermediate scale patterns may be hidden by the domain-wide trends.

**Table S.2.** Large-scale linear trends of entirely-filtered radar and WRF precipitation patterns on the local domain (Figure 1 in the main text). *Orientation* gives the direction of the slope and *Intensity* the strength of inclination. 0° would indicate a slope pointing toward the East. WRF snow precipitation is from simulations with weak terrain smoothing (Sect. 2.1 in the main text).

| | 31 January 2016 Orientation | Intensity | 4 February 2016 Orientation | Intensity | 5 March 2016 Orientation | Intensity |
|---|---|---|---|---|---|---|
| Radar entirely filtered | 68.5° | 0.14 | 150.5° | 0.03 | -135.4° | 0.04 |
| WRF precip. at 2830 m asl | 22.6° | 0.26 | 5.8° | 0.24 | -79.6° | 0.19 |
| WRF total ground precip. | 30.6° | 0.32 | 24.2° | 0.24 | -67.6° | 0.21 |

[Figure]

**Figure S.1.** Domain-wide 24 h precipitation statistics for the local domain (300 m resolution, Figure 1 in the main text) for the three precipitation events on 31 January 2016, 4 February 2016 and 5 March 2016. Gray colors show entirely-filtered radar precipitation. WRF precipitation at 2830 m above sea level (m asl) for simulations with weak terrain smoothing (Sect. 2.1 in the main text) and strong terrain smoothing are given in blue and violet, respectively. Orange (red) shows boxplots of WRF total ground precipitation for weak (strong) terrain smoothing. Radar precipitation and WRF precipitation at 2830 m asl are masked (as shown in Figure 4 in the main text).

[Figure]

**Figure S.2.** Normalized variograms of detrended snow precipitation for the precipitation events on a) 31 January 2016, b) 4 February 2016 and c) 5 March 2016 for the local domain (Figure 1 in the main text). Variograms are given for partly-filtered (red) and entirely-filtered (orange) radar snow precipitation, WRF snow precipitation at 2830 m above sea level (m asl, blue) and WRF total ground precipitation (violet). WRF precipitation is from simulations with weak terrain smoothing (Sect. 2.1 in the main text). All precipitation fields are masked.

[Figure]

**Figure S.3.** Normalized variograms of the snow precipitation events on a) 31 January 2016, b) 4 February 2016 and c) 5 March 2016 for the local domain (Figure 1 in the main text). Variograms are given for entirely-filtered radar snow precipitation (orange), WRF snow precipitation at 2830 m above sea level (m asl, blue) and WRF total ground precipitation (violet). Additionally, variograms are given for real topography (based on dhm25 © 2018 swisstopo (5740 000 000), black) and WRF topography (gray). WRF topography and precipitation are from simulations with weak terrain smoothing (Sect. 2.1 in the main text). All precipitation fields are masked.

---

## Author Response (AR2)

**Editor Decision: Publish subject to minor revisions (review by editor)** (24 Jul 2018) by Ross Brown

Comments to the Author:

Dear Authors, I have the comments back on your revised m/s which was reviewed very favorably by 2 of the 3 reviewers. One reviewer (report 3 included below) still has concerns about the way you have formed your conclusions, and I ask you to provide a detailed response for my consideration as this has a clear impact on the take home messages from your paper. Another reviewer noted a number of typos so please check these also.

Best regards, Ross Brown (ed.)

Dear Ross Brown,

Thank you, Heini Wernli and the two anonymous reviewers for the evaluation of our manuscript. We have addressed the questions and comments as pointed out in report #3 below and hope that we were able to clarify the statements. Point-by-point answers are given below including our changes in the manuscript. Changes in the manuscript are put in quotation marks, page and line numbers refer to the track-changes version of the manuscript below. We gave our best to correct the typos.

On behalf of all authors,
Franziska Gerber

**Report 3:**

General considerations

The authors have satisfactorily addressed most of the comments of the first review. However, they still have a hard time to give up the idea that the forcing model should be the reason for the bad performance of their model chain. I have reformulated my former major comment 1 and still call it 'major'. Apart from this I think there are only a few minor issues to be addressed.

Major comments

1) Still: the authors try to 'explain' WRF offsets (Section 3.1, point inter-comparison) by 'wrong COMO input'. It is (still) argued that these offsets were (partly) 'due to offsets in the COSMO-2 input' (p11, l. 3/5; p.13, l. 32). As I pointed out in my first review, this is a very weak argument and must completely be omitted. Rather, the performance relative to COSMO – or to the coarser WRF simulations - should be used in the discussion of possible reasons. Fact is that for none of the variables, there is a clear trend to obtain better overall correspondence to the observations when increasing the model resolution. Sometimes, (e.g., 31.1., Dischma moraine, temperature) indeed the 50m grid spacing is 'a little better' (and it seems to be systematic from coarse to fine), sometimes (e.g., 5.3., Dischma moraine, RH) COSMO overestimates, and this is reduced the finer the resolution becomes (leading to an underestimation at 50 m grid spacing…), sometimes (31.1., 4.2., Moraine, wind speed) COSMO overestimates and the high resolution makes it worse, and sometimes (31.1., 4.2., FLU2), wind speed is astonishingly good in COSMO – and higher resolution makes it (much) worse. So, what one, first of all, may conclude is, that it NOT the input that determines the high-resolution performance. Second, one may state that T, RH and WD are 'reasonably wrong' (in all resolutions) and might be explainable to some degree (as the authors do). Wind speed, on the other hand, is sometimes unreasonably wrong and therefore deserves a somewhat more in depth discussion. Again (as in my first review), it first should be stated that only if the simulations are adequate (spin up time, distance from the (upwind) boundary, …) it makes sense to investigate the

discrepancies (the authors do this quite briefly on p.13, l. 33). Second, it should be stated that the improvement is not systematic (see above), i.e. not 'better with finer resolution'. Rather, the varying performance seems to indicate that we are dealing here with compensating errors. Another aspect which is clearly evident (and it also shines through in the 'explanations' by the authors) is that the scales of the model and the observations do not match. It is rather difficult to address this issue in the observations, but what can be done is to estimate the area of influence (footprint) to judge how this length scale relates to the grid resolution. As for the model, I recently came across an interesting 'grid point ensemble' approach (Goger et al. 2018), which could also be tried here.

I think the authors should abstain from the (quite often observed) desire to 'have a good model performance' (p14, l.8). Very often, the bad performances are much more conclusive and advance our knowledge much more than calling a bad result good. What we can learn from this exercise is that finer resolution does not necessarily lead to better 'point-performance' in complex terrain (certainly not if the necessary conditions are not respected); maybe another outcome can be what the scales of observations and simulations are, and finally that wind speed is much more challenging than other variables (since one can be in a completely different flow regime with the measurement than with the model) – and additional research is necessary to address these issues.

As the reviewer mentions, we get some very diverse results sometimes improving the results by higher resolutions sometimes worsening them. Based on the reviewer comments we try to point this out more precisely. Concerning the COSMO-2 input, we agree that this is an unlikely reason. We decided to keep it as part of the discussion mentioning that if the large-scale gradient in the input model would be inadequate, WRF may not correct for this error. However, we point out that for our simulations this is an unlikely cause given the divers agreement of COSMO-2 compared to station measurements (overestimation of wind speeds by COSMO for station Dischma Moraine but good estimation of wind speeds by COSMO for station FLU2 on 31 Jan and 4 Feb 2016). Therefore, we removed the discussion about COSMO for the single parameters (P. 12ff) but add a short paragraph after discussing wind speeds:

P. 14, l. 15-21: "While the model is designed such that it develops independently (given its fetch distances and spin-up times, Sect. 2.1), a poor representation of the large-scale gradients in the COSMO–2 input might not be corrected by COSMO–WRF. The investigated variables do not show a consistent signal of improvement nor a consistent signal of worsening with higher resolutions and with respect to the COSMO–2 input. Similarly but not consistently in phase with COSMO–WRF, COSMO–2 shows a good agreement with station measurements for certain cases, while it shows worse performance for other cases. Given these inconsistencies between cases and for COSMO–2 input a poor representation of the large-scale gradient in COSMO–2 is an unlikely reason for a bad model performance."

Concerning the 'grid point ensemble' method to evaluate the point performance, we agree upon this being an interesting method. While Goger et al., 2018 use the grid point closest to the station and its 8 neighbors, we linearly interpolate between the four nearest neighbors of the station, thereby taking into account some of the variability around the station. Indeed, this approach should be refined and an in depth study of the area of influence should be performed in upcoming studies. We more precisely point out in the methods section (Sect. 2.1) that the variables have been interpolated between the four nearest neighbors of the station and we mention the study by Goger et al., 2018 as an alternative approach:

P. 6, l. 34: "…, using model output of the four grid points surrounding the station, …"
P. 6. l. 35ff: "Alternatively, the eight neighbors of the grid point closest to the station could be used (Goger et al., 2018)."

In addition, we adapted the conclusion pointing out the speed-up effects and the lack of small-scale terrain features are not the only causes for overestimated wind speeds. We agree, and tried to more precisely point this out, that there are likely other (unknown) reasons for these deviations and that additional work is needed to address these issues (see also comment on minor comment #8):

P. 24, l. 31-P.25, l. 2: "… but tends to (strongly) overestimate near-surface wind speeds. This may be due to many reasons from an overestimation of speed-up effects on the windward side of mountain ridges to an underrepresentation of small terrain features. Additional reasons are likely but remain unknown and future work is needed to address these issues."

Minor comments
P3, l.26 the question should read: 'To what degree is snow precipitation variability represented by…'
We thank the reviewer for this suggestion and changed the question to "To what degree is snow precipitation variability represented by …" (P.3, l. 26-27).

P7, l.4 '…are used for the correction': I still consider this inappropriate to be called 'a correction'. I think it would be appropriate to say that the meteorological information (wind speed, friction velocity) from the first model level is used for the interpolation to the actual height of observation.
We agree that correction may not be appropriate but we rather use the term "extrapolation" instead of "interpolation". We changed "… are used for the correction" to "… are used for the extrapolation" (P.7, l. 6).

P9, l. 24/26 is the semivariance the 'gamma' (as on l. 24) or the 'gamma hat' as in the equation?
Thank you for pointing this out. The "gamma" should actually read "gamma hat", which we corrected.

P10, l.25 if the sentence introduces all variables, the reference to Fig. 2a-c, Fig 3a-c is wrong, since these panels only show temperature.
Indeed this sentence introduces all variables and we changed "Fig. 2a-c, Fig. 3a-c" to "Fig. 2, 3". We added "Fig. 2a-c, Fig. 3a-c" on page 11 line 2, where we only talk about temperature.

P13, l. 24 'for the two events…': the same is true for the third event, for about 6 hours around noon at Flu2….
We added "… and for the precipitation event on 5 March 2016 at stations FLU2 for about 6h around noon." (P. 13, l. 31) and changed the flow of arguments (P. 13ff).

P17, l. 8 this supports the hypothesis (no confirmation at all)
We changed "confirms" to "supports" (P. 17, l. 26).

Tab 2, Fig 6, 7, caption: WRF simulation at which grid spacing?
The figure captions state that the data are shown for the "regional domain", which has a horizontal grid spacing of 450 m. To help the reader we added the specification of the horizontal grid spacing in the captions of Tab 2, Fig 6 and Fig 7 as: "… the regional domain (horizontal grid spacing of 450 m, Fig 1) …".

P25, l.6 this is likely not a good summary of why the wind speed is overestimated (but not always), see major comment 1.

We agree that the summarized reasons are not exclusive and thus added a statement pointing out that additional reasons remain unknown.

"… but tends to (strongly) overestimate near-surface wind speeds. This may be due to many reasons from an overestimation of speed-up effects on the windward side of mountain ridges to an underrepresentation of small terrain features. Additional reasons are likely but remain unknown and future work is needed to address these issues." (P. 24, l. 31 – P. 25, l. 2)

P25, l.15 'likely due to the high model resolution': this is a quite misleading argument. The high model resolution is more realistic. Thus, when the 'high model resolution' produces too much precipitation, this indicates that the lower model resolutions were likely wrongly tuned to overcome an underestimation due to coarse model resolution.

Indeed the higher model resolution should be more realistic. However, model physics have been developed for coarser grid spacing. It has been reported previously that an overestimation, especially over steep and complex terrain may occur for high resolution simulations (e.g. Silverman et al., 2013). This has been attributed to different causes one of which was that higher and steeper mountains in high resolution simulations may produce "too much" precipitation likely because model physics are developed for simulations with coarser resolutions. This is consistent with our findings and given the very high resolution in our simulations (by far higher than resolutions conventionally used) a reasonable cause for an overestimation of precipitation in the model. We decided to keep the argument but emasculate by mentioning that this is one potential reason among others. "
[revised manuscript text omitted]